# DensBO: Dynamic Ensembling of Surrogate Models for Hyperparameter Optimisation

## Abstract

Hyperparameter optimisation (HPO) of machine learning models is crucial for achieving optimal performance for different tasks. Surrogate-based optimisation techniques, such as Bayesian optimisation (BO), have been successfully applied to tackle this problem. BO is subject to different design choices of its components. In particular, depending on the nature and the size of the search space, the choice of the surrogate model has a substantial impact on the overall performance of BO. Surrogate models in BO approximate the function to optimise and guide the search towards promising regions by predicting the function value for different solution candidates. Combining different machine learning (ML) models is known to lead to performance gains, *e.g.*, in different prediction tasks. To this end, we propose a novel dynamic approach to ensemble surrogate models in the BO pipeline, leveraging the complementary powers of different surrogate models at different stages of the optimisation process. We empirically evaluate our method on numerous benchmarks and demonstrate its advantage compared to state-of-the-art single-surrogate BO baselines. We highlight the usefulness of our approach in finding good hyperparameter configurations in mixed (numerical and categorical) search spaces for a wide range of problems.

## 1 Introduction

The performance of machine learning (ML) and deep learning (DL) models crucially depends on how their hyperparameters are tuned (Lavesson & Davidsson, 2006; Bischl et al., 2023). Tuning the hyperparameters in order to achieve peak performance of the model on a specific task is challenging even for experts. This process is often addressed through trial-and-error methods, requiring significant effort and resources. Hyperparameter optimisation (HPO) techniques alleviate this burden by automatically searching for well-performing hyperparameter configurations, removing the need for human intervention (Snoek et al., 2012). While automated HPO techniques have shown great potential for classical ML models (Snoek et al., 2012; Feurer et al., 2022), they are not as easily applicable to more complex ML and DL domains, where evaluating a single configuration of a model can be very expensive (Brown et al., 2020). Moreover, due to the lack of access to an explicit problem formulation, HPO is handled as a black-box problem. Consequently, all automated HPO techniques rely on the only available information about the problem, *i.e.*, evaluating the quality of candidate configurations, to steer the search towards the most promising regions of the search space and a good estimate of the global optimum.

Bayesian optimisation (BO) (Močkus, 2012; Garnett, 2023; Frazier, 2018) is a surrogate-based, sample-efficient approach for global optimisation of expensive-to-evaluate black-box problems. BO is particularly well-suited for settings with a very limited budget of available evaluations (relative to the size of the search space), such as HPO. Various BO-based HPO techniques have been developed and successfully applied to this end; a prominent example is HEBO (Cowen-Rivers et al., 2022), the winner of the NeurIPS BBO competition (Turner et al., 2020). One of the key components of BO is a probabilistic *surrogate model*, which is built on an initial sample of solution candidates to approximate the objective function while also quantifying its uncertainty. At each iteration, a new solution candidate is selected to be evaluated next by means of maximising an acquisition function

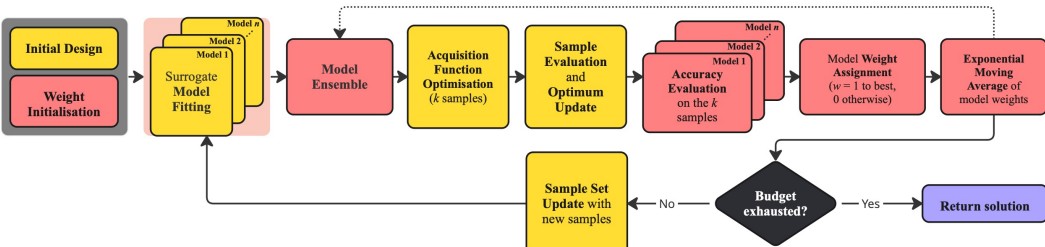

Figure 1: Bayesian optimisation pipeline with dynamic ensembling of surrogate models. In red, the blocks for the ensembling strategy that have been plugged into the standard BO pipeline. We start by sampling an initial set of hyperparameter configurations and initialising the weights for surrogate models used to construct the ensemble. Within the BO loop, all surrogates are separately fitted to the past observations and then combined in the weighted ensemble. The acquisition function is derived from the model ensemble and optimised to determine the next points to sample. The accuracy of the surrogate models is then evaluated on the newly sampled points and they are assigned new weights. Finally, the weighting scheme for the next iteration is obtained via an exponential moving average between the old and new weights, which is parameterised by a smoothing factor in order to determine how much historical information is retained.

defined on the surrogate. The surrogate is then iteratively refined with newly observed solution candidates, until the total budget of available evaluations has been exhausted.

Fitting the surrogate model to the observed data is a prediction task that often involves probabilistic models. BO then uses these probabilistic predictions of function values across the search space to guide the search towards a good estimate of the global optimal solution. The choice of the surrogate model thus strongly affects BO performance and is often linked to the dimensionality of the problem at hand and the nature of its search space. For continuous search spaces, Gaussian processes (GPs) (Rasmussen & Williams, 2006) are the most widely adopted surrogate model and tend to be particularly effective on low-dimensional problems, involving up to approximately 20 variables (Eggensperger et al., 2013). On the other hand, random forests (RFs) (Breiman, 2001) as surrogate models natively support discrete and conditional search spaces, and tend to excel in higher problem dimensionalities, where GPs generally do not work well (Eggensperger et al., 2013; Jenatton et al., 2017; Li et al., 2017). For complex HPO tasks in mixed domains, with numerical, ordinal and categorical hyperparameters, it is highly desirable to leverage the complementary strengths of inherently different surrogate models. One natural way to achieve this is via ensemble methods, which have in practice demonstrated their versatility in the context of BO-based HPO techniques for treating a wide range of heterogeneous problems (Turner et al., 2020; Hoffman et al., 2011).

We propose a novel approach to enhance BO-based HPO performance by dynamically ensembling surrogate models during the optimisation process. Our approach, which we dub `DensBO`, is based on a weighted combination of multiple surrogate models, assigning the largest weight to the surrogate model with the highest accuracy on newly observed points in each iteration and updating the weights via exponential moving average (see Figure 1 for details).

We present, for the first time, a dynamic ensembling approach for surrogate models in the context of HPO. This poses a significant challenge, as there is a trade-off between the time needed for finding the best hyperparameter configuration and the time complexity of target function evaluations (*i.e.*, if the optimiser requires more time to determine the next configurations to sample, there is naturally less time available for evaluating the target function). In the context of HPO, evaluating the target function means evaluating the performance of a given ML model for a given hyperparameter configuration on a given dataset. Ensembling approaches in general require training and querying more than one surrogate model, which leads to more time required for the optimisation phase.

We assess the effectiveness of the `DensBO` approach on various HPO tasks involving numerous datasets and machine learning models. We compare it to several single-surrogate-based BO and a simple, static ensemble, and we show that our dynamic ensembling method outperforms all of them

on both cheap and expensive functions. We provide the code, instructions for reproducibility, as well as all figures in the Appendix C.

The remainder of this paper is organised as follows: in Section 2, we define the HPO problem, describe how BO operates, and position our approach with respect to related work on dynamically adapting BO components and using ensembles for enhancing BO performance. Section 3 introduces our methodology for dynamic ensembling of surrogate models. In Section 4, we provide the technical details on the experimental setup and describe the benchmarks and baselines chosen for our empirical analysis. We present results and critically discuss them in Section 5. Finally, in Section 6, we provide concluding remarks and outline directions for future research.

## 2 BACKGROUND AND RELATED WORK

In this section, we define the HPO problem and describe the working mechanisms of the BO framework. We also cover related work on dynamic design choices related to BO's components, as well as using ensembles in BO.

### 2.1 HYPERPARAMETER OPTIMISATION

Let $\mathcal{A}$ be a learning algorithm with $n$ hyperparameters, $\Lambda_i$ the domain of the $i$-th hyperparameter, and $\mathbf{\Lambda} = \Lambda_1 \times \Lambda_2 \times \ldots \Lambda_n$ the overall hyperparameter configuration space. We denote a hyperparameter configuration by $\lambda \in \mathbf{\Lambda}$, and the algorithm $\mathcal{A}$ with its hyperparameters instantiated to $\lambda$ by $\mathcal{A}_\lambda$. Given a dataset $D$, the objective of HPO is to find a hyperparameter configuration $\lambda^*$ that minimises the loss $\mathcal{L}$ of a model fitted by algorithm $\mathcal{A}$ with hyperparameters $\lambda$ on training data $D_{train}$, and evaluated on validation data $D_{validate}$, for a given loss function $\mathcal{L}$, *i.e.*,

$$\lambda^* \in \operatorname*{arg\,min}_{\lambda \in \mathbf{\Lambda}} \mathcal{L}(\mathcal{A}_\lambda, D_{train}, D_{validate}) = \operatorname*{arg\,min}_{\lambda \in \mathbf{\Lambda}} c(\lambda) \tag{1}$$

Here, $c(\lambda)$ is a shorthand for the estimated loss function when $\mathcal{A}_\lambda$ and $D$ are fixed. Note that $c(\lambda)$ is a black-box function, without a closed-form mathematical expression nor analytic gradient information.

### 2.2 BAYESIAN OPTIMISATION

Bayesian optimisation (BO) (Močkus, 2012; Frazier, 2018; Garnett, 2023) is a family of surrogate-based algorithms for efficient global optimisation of black-box problems. A typical BO pipeline consists of three main modules: an *initial design*, *i.e.*, in HPO, a set of hyperparameter configuration candidates $\boldsymbol{\lambda} = (\lambda^{(1)}, \ldots, \lambda^{(r)})$ and their evaluations $c(\boldsymbol{\lambda}) = (c(\lambda^{(1)}), \ldots, c(\lambda^{(r)}))$; a *surrogate model* (fitted to the initial observations) that returns an approximation $\hat{c}(\lambda)$ of the unknown loss function $c(\lambda)$ while capturing the uncertainty in the prediction $\hat{\sigma}(\lambda)$ on unobserved points in the search space; and an *acquisition function*, which is optimised to suggest solution candidates to be evaluated next, usually balancing exploration and exploitation of the search space. To approximate the expensive objective function, BO typically employs a Gaussian process (GP) model as the surrogate. The Gaussian process model defines a distribution over functions on the configuration space $c(\lambda) \sim \mathcal{GP}_c(\mu(\lambda), k(\lambda, \lambda'))$, where $\mu(\cdot)$ is a mean function and $k(\cdot, \cdot)$ is a covariance function. If we consider an observation model $y_i = c(\lambda^{(i)}) + \varepsilon_i$ with normally distributed noise, $\varepsilon_i = \mathcal{N}(0, \sigma_\varepsilon^2)$, the predicted value by the Gaussian process model at one unknown configuration $\lambda$ will also follow a normal distribution with mean $\mu(\lambda) = K_{\lambda, \boldsymbol{\lambda}}(K_{\boldsymbol{\lambda}, \boldsymbol{\lambda}} + \sigma_\varepsilon^2 I)^{-1} \boldsymbol{y}$ and variance $\sigma^2(\lambda) = k(\lambda, \lambda) - K_{\lambda, \boldsymbol{\lambda}}(K_{\boldsymbol{\lambda}, \boldsymbol{\lambda}} + \sigma_\varepsilon^2 I)^{-1} K_{\boldsymbol{\lambda}, \lambda}$, where $\boldsymbol{y} = (y_1, \ldots, y_r)$ is the vector of observations, $K_{\boldsymbol{\lambda}, \boldsymbol{\lambda}} = [k(\lambda^{(i)}, \lambda^{(j)})]_{\lambda^{(i)}, \lambda^{(j)} \in \boldsymbol{\lambda}}$ is the covariance matrix, and $K_{\boldsymbol{\lambda}, \boldsymbol{\lambda}} = [k(\lambda^{(i)}, \lambda)]_{\lambda^{(i)} \in \boldsymbol{\lambda}}$ is the correlation vector for all samples. GPs as a surrogate model inherently provide both the mean and the variance vector. However, GPs are not the only surrogate model used in BO. Another common choice for a surrogate model are tree-based models, such as random forests. Tree-based models traditionally predict only the mean of the given data (*e.g.*, in a regression setting). In this case, the variance is defined based on the variance of the predictions of the leaves (Hutter et al., 2011). We calculate the mean $\mu(\lambda)$ and variance $\sigma(\lambda)$ for a set of trees $T$ as follows:

$$\mu(\lambda) = \frac{1}{|T|} \cdot \sum_{t \in T} t(\lambda), \tag{2}$$

$$\sigma(\lambda) = \frac{1}{|T|} \cdot \sum_{t \in T} (t(\lambda) - \mu(\lambda))^2 \, , \tag{3}$$

where $t(\lambda)$ is the prediction of a tree $t \in T$.

BO proceeds iteratively until a termination criterion is met. In each iteration, it optimises the acquisition function by repeatedly querying the surrogate model to generate a pool of solution candidates. It then evaluates the most high-potential solutions (*i.e.*, the solutions that maximise the acquisition function) from this pool, refines the surrogate model based on the new observations, and updates the optimum if the new point improves upon the true function value of the best observation so far. Among the many variants of BO from the literature, in this work, we focus on the state-of-the-art method for HPO to empirically evaluate our ensembling method: HEBO (Cowen-Rivers et al., 2022).

## 2.3 DYNAMIC COMPONENT SELECTION IN BAYESIAN OPTIMISATION

As a modular framework, BO performance is highly sensitive to design choices of its modules. Different sampling strategies for the initial design, such as Latin hypercube sampling (McKay et al., 2000), low-discrepancy sequences (*e.g.*, Sobol (Antonov & Saleev, 1979)) or random uniform sampling; different surrogate models, such as GPs or RFs; and different acquisition functions (AFs), such as expected improvement (EI) (Močkus, 1975), probability of improvement (PI) (Kushner, 1964) or upper confidence bound (UCB) (Forrester et al., 2008) – all affect the overall BO performance to various degrees (Bossek et al., 2020; Lindauer et al., 2019; Cowen-Rivers et al., 2022). Despite few works showing the potential of automated selection of components (Ben Salem & Tomaso, 2018; Benjamins et al., 2022a;b), the settings for each component are typically chosen by practitioners beforehand depending on the desired use-case, and are fixed for the entire optimisation procedure. However, there have been efforts to show that the dynamic choices of BO modules lead to performance gains across multiple contexts. Prior attempts to investigate the dynamic adjustment of AFs include works on mixed AF strategies, *e.g.*, a self-adjusting AF approach to balance the exploration-exploitation trade-off (Benjamins et al., 2023), an online multi-armed bandit strategy on a portfolio of AFs (Hoffman et al., 2011), or an online update of weights in a portfolio of AFs (Kandasamy et al., 2020). When it comes to dynamic adjustment of surrogate models, several directions have been investigated, notably an online selection of surrogate models based on their ranking in each BO iteration (Bagheri et al., 2016), adaptive global surrogate modelling via genetic algorithm-driven sampling (Gorissen et al., 2009), or adaptive combining of surrogates based on crowding distance trust regions (Zhang et al., 2012). It has also been shown that dynamic component selection in general is beneficial in terms of performance in other related areas, *e.g.*, in algorithm configuration (Biedenkapp et al., 2020), evolutionary computation (Karafotias et al., 2015; Doerr & Doerr, 2020), planning (Speck et al., 2021), and deep learning (Adriaensen et al., 2022).

## 2.4 ENSEMBLES IN BAYESIAN OPTIMISATION

Ensembles of ML models have been shown to outperform single models for a wide range of use-cases (Sagi & Rokach, 2018; Opitz & Maclin, 1999; Rokach, 2010; Dong et al., 2020). Consequently, using ensembles in the context of BO is not a new idea. A series of works has demonstrated the advantage of using ensembles, most notably in engineering (Jiang et al., 2020; Zhou et al., 2011). Various ensembling strategies have been investigated, *e.g.*, optimal weighting of surrogates trained on existing observations (Hanse et al., 2022) or on extracted features (Guo et al., 2019), or optimising multiple AFs on multiple surrogates and combining them accordingly (Huang et al., 2022; Beaucaire et al., 2019). However, all these works consider a static (*i.e.*, global) ensemble construction which is then used throughout the entire optimisation. In contrast to this, our method operates in a dynamic fashion, tracking the accuracy of surrogates and adjusting the ensemble on-the-fly.

Dynamic model ensembles have been used with time-series data (Liu, 2023), or for approximating both high- and low-fidelity data during the BO procedure by combining two GP regression models (Liu, 2020); they have also been considered in a non-BO context, *e.g.*, for neural decoding in brain-computer interfaces (Qi et al., 2019). However, these approaches substantially differ from our proposed methodology, as none of them considers the use of exponential moving average to capture the history of model performance, nor a weighting scheme that is based on the accuracy on the newly sampled points of the surrogate models trained on past observations. Furthermore,

we consider ensembles of regression models from inherently different families. To the best of our knowledge, dynamic surrogate ensembles have so far not been applied in the context of HPO. We thus not only consider a new use-case, but an entirely different challenge compared to other real-world applications, due to a key trade-off between the budget for evaluating the target function and the budget for finding the optimal hyperparameter configuration.

## 3 METHODOLOGY

Our dynamic ensembling approach works as follows. The BO procedure is launched with an initial set of hyperparameter configurations. We then initialise the weights and assign them to the models used to construct the ensemble. We denote the initial weight for the surrogate model $m$ as $w_{0,m}$. Then, the BO loop begins. We train all surrogate models separately on all the available samples and create a weighted ensemble. At each iteration $t$, the weighted ensemble is a convex combination of the surrogate models, with coefficients defined as the normalised model weights of that iteration, *i.e.,*:

$$\hat{w}_{t,m} = \frac{w_{t,m}}{\sum_{j \in M} w_{t,j}}, \tag{4}$$

where $M$ is the set of available surrogate models and $w_{t,m}$ is the weight of model $m$ at iteration $t$. This way we make sure that $\sum_{m \in M} \hat{w}_{t,m} = 1$ with $0 \leq \hat{w}_{t,m} \leq 1$. The ensemble can then predict the $\mu_{\text{ens}}(\lambda)$ and $\sigma_{\text{ens}}(\lambda)$ of the performance for an unobserved configuration $\lambda$ by using a weighted average of the individual model predictions $\mu_m(\lambda)$ and their standard deviations $\sigma_m(\lambda)$:

$$\mu_{\text{ens}}(\lambda) = \sum_{m \in M} \hat{w}_{t,m} \cdot \mu_m(\lambda), \tag{5}$$

$$\sigma_{\text{ens}}(\lambda) = \sum_{m \in M} \hat{w}_{t,m} \cdot \sigma_m(\lambda). \tag{6}$$

We note that, in preliminary experiments, we investigated a variation of our ensembling method that applies different weighting schemes for the mean and the variance. However, this did not yield sufficient improvements to justify a more complex method definition. Further details can be found in Appendix H.

In the next stages, at each iteration $t$, we optimise the acquisition function and select the candidates to sample $\lambda_t^{(1)}, \lambda_t^{(2)}, \ldots, \lambda_t^{(k)}$, as is typically done in BO. We then evaluate the accuracy of the different surrogate models using mean squared error (MSE) on the newly sampled points. Based on the calculated MSE, we determine the new weights $\bar{w}_m$ for each model as follows:

$$\bar{w}_{t,m} = \begin{cases} 1 & \text{if } \text{MSE}(\mu_m(\lambda_t), c(\lambda_t)) = \min_{l \in M} \text{MSE}(\mu_l(\lambda_t), c(\lambda_t)), \\ 0 & \text{otherwise}, \end{cases} \tag{7}$$

where $\lambda_t = \{\lambda_t^{(1)}, \lambda_t^{(2)}, \ldots, \lambda_t^{(k)}\}$. Therefore, in Equation 7, we assign a weight of $\bar{w}_{t,m} = 1$ to the model that has the lowest MSE on the $k$ new points $\lambda_t$ sampled at iteration $t$. Then, for each model $m$, we use an exponential moving average between the previously calculated weights and the new weight:

$$w_{t+1,m} = (1 - \alpha) \cdot w_{t,m} + \alpha \cdot \bar{w}_{t,m}, \tag{8}$$

We use the exponential moving average in the weighting scheme to account for the history of the surrogate model performances, which is crucial for performing continual ensembling rather than only model selection at each iteration. However, we also want to be responsive to the recent tendencies of the acquisition function, and thus accordingly reward the models that are accurate in the regions of the search space which are favoured by the acquisition function.

We summarise our method in the pseudo-code description provided in Algorithm 1.

**Initialisation (Lines 1–3).** An initial set of $r$ hyperparameter configurations $\{\lambda_1, \ldots, \lambda_r\}$ is generated based on the sampling scheme of the chosen BO framework and then evaluated. The weights $w_{0,m}$ defining the model ensemble are also initialised. The iterative phase of BO starts and is run until the budget is exhausted.

**Model fitting (Lines 5–6).** All models used to construct the ensemble are fitted to the data. Then, the model ensemble is computed as a weighted average of the single models.

**Augment dataset with new points (Lines 7–9).** The acquisition function $AF$ of the chosen BO framework is optimised on the model defined by $\mu_{\text{ens}}, \sigma_{\text{ens}}$ to generate $k$ new solution candidates to improve the current best solution. The target function is evaluated on the new candidates, and the problem dataset is augmented with these new points.

**Update weights (Lines 10–11).** New model weights are computed based on the accuracy of the single models evaluated on the newly sampled solutions and weight history.

**Return best configuration (Line 13).** The best found hyperparameter configuration $\lambda^*$ is returned as the optimal solution.

Note that, since our method is a plug-in for a generic BO framework, some of the steps (initial sample generation and acquisition function optimisation) depend on the specific BO framework.

---

**Algorithm 1** `DensBO`: Dynamic Ensembling in BO

---

**Input**: total budget $b$, size of newly sampled batch $k$, initial sample size $r$, loss function $c$, portfolio of surrogate models $M$, acquisition function $AF$,

1: Initialise $\boldsymbol{\lambda}$ with $r$ randomly sampled points
2: Evaluate initial samples: $\boldsymbol{C} \leftarrow \{c(\lambda) \mid \lambda \in \boldsymbol{\lambda}\}$
3: Initialise model weights $w_{0,m}$
4: **while** budget is not exhausted **do**
5:     Fit models to data: $\{(\mu_m, \sigma_m) \mid m \in M\} \leftarrow \{\text{fit}(m, \boldsymbol{\lambda}) \mid m \in M\}$
6:     Generate model ensemble $(\mu_{\text{ens}}, \sigma_{\text{ens}})$ according to Equations 5 and 6
7:     Optimise acquisition function: $\lambda_t = \{\lambda_t^{(1)}, \ldots, \lambda_t^{(k)}\} \leftarrow \arg\max AF(\lambda, \mu_{\text{ens}}, \sigma_{\text{ens}})$
8:     Evaluate new candidates: $c_t = \{c_t^{(1)}, \ldots, c_t^{(k)}\} \leftarrow \{c(\lambda) \mid \lambda \in \lambda_t\}$
9:     Augment dataset: $\boldsymbol{\lambda} \leftarrow \boldsymbol{\lambda} \cup \lambda_t$, $\mathbf{C} \leftarrow \mathbf{C} \cup c_t$
10:    Calculate new model weights: $\bar{w}_{t,m}$ according to Equation 7
11:    Update weights $w_{t,m}$ according to Equation 8
12: **end while**
13: **Return** best configuration $\lambda^* \in \arg\min_{\lambda \in \boldsymbol{\lambda}} \mathbf{C}$

---

## 4 EXPERIMENTAL SETUP

We conducted a range of experiments to assess the performance of `DensBO` in various HPO settings compared to single-surrogate baselines and a static ensemble. In particular, we investigated how `DensBO` performs in light of the trade-off between the time for evaluating the target function and the time for finding the optimal hyperparameter configuration. We implemented our method in HEBO (Cowen-Rivers et al., 2022), a state-of-the-art HPO framework (Eggensperger et al., 2021). HEBO is a BO-based optimiser that includes several advancements to improve performance. It applies a power transformation to the performance data and the Kumaraswamy transformation to the input data to tackle heteroscedasticity and non-stationarity. Cowen-Rivers et al. (2022) showed that these transformations improve the performance of Gaussian processes on performance data. HEBO samples new solution candidates by maximising a multi-objective acquisition function that consists of EI, PI, and upper confidence bound (UCB) (Forrester et al., 2008).

We considered four classes of surrogate models to construct the ensemble: Gaussian processes and three tree-based models, namely random forest (RF) (Breiman, 2001), extremely randomised trees (ET) (Geurts et al., 2006) and gradient boosting (GB) (Friedman, 2001). For GP, we used its native implementation from HEBO. For tree-based models, we used the `scikit-learn` implementation of RF, ET and GB. For details on these methods, refer to Section C.3.

As GPs are preferable for scenarios with continuous hyperparameters and RFs better handle discrete (and even mixed and conditional) domains (Eggensperger et al., 2013), we initialised the weights in the following manner: if the problem instance contains only continuous hyperparameters, we assign a weight of 1 to Gaussian process ($w_{0,GP} = 1$) and 0 to all other surrogate models; otherwise, we assign a weight of 1 to RF ($w_{0,RF} = 1$) and 0 to all other surrogate models.

We used four values of the smoothing factor $\alpha$ of the exponential moving average for the dynamic ensemble construction: $\alpha \in \{0.1, 0.5, 0.9, 1.0\}$. The lower the $\alpha$, the lower the impact of the new weights in each iteration, *i.e.*, the higher the impact of the history of the weights. This particular

choice thus reflects low, medium and high memory and a selection-only mode when $\alpha = 1$. In the latter case, we select only one model – the one with the highest accuracy on the newly sampled configurations from the last iteration – rather than construct a weighted ensemble.

In each iteration, the models with null weights are dynamically pruned, *i.e.*, not considered at inference time. In the case of $\alpha = 1$, only the model achieving the highest accuracy on the newly sampled solutions is assigned a weight of $w = 1$, while all other models receive $w = 0$. Thus, $\alpha = 1$ performs model selection rather than ensembling, as no historical information is used. The $\alpha = 1$ scenario leads to the shortest running time, due to the fact that in acquisition function optimisation having fewer models with non-zero weights reduces inference time, since fewer models need to be accessed altogether.

## 4.1 BENCHMARKS AND BASELINES

We empirically evaluated our approach on two benchmark suites, YAHPO Gym (Pfisterer et al., 2022) and JAHS-Bench-201 (Bansal et al., 2022). YAHPO Gym is a surrogate-based benchmark for hyperparameter optimisation, consisting of 15 scenarios (*i.e.*, machine learning pipelines and their configuration space) on various datasets. In particular, it contains LCBench (Zimmer et al., 2021), which is used to optimise neural networks on tabular data, combined algorithm selection and hyperparameter optimisation (CASH) on OpenML datasets (Binder et al., 2020; Falkner et al., 2018), as well as the NAS Bench 301 (Zela et al., 2022). From YAHPO Gym, we used all 856 available instances. JAHS-Bench-201 is a surrogate-based benchmark for optimisation of the architecture and hyperparameters of convolutional neural network on image datasets. From JAHS, we used all three available instances, each optimising the architecture for a different dataset. Both YAHPO and JAHS contain mixed-type configuration spaces. For further experimental and implementation details, refer to Appendix C.

We used single-surrogate BO variants within HEBO with each of the surrogate models considered (GP, RF, ET, and GB) as baselines to compare our dynamic ensembling approach. Furthermore, we compared against a simple static ensemble which always assigns an equal weight to all models.

## 4.2 EXPERIMENTAL PROTOCOL

We used the native implementations of the initial design sampling and the acquisition function in HEBO. We set the size of the initial sample to 8 for all experiments. Each method ran with 51 different random seeds and with a total budget of $100\times$ mean evaluation time, similarly to Eggensperger et al. (2021). For more details on the evaluation time, refer to Appendix D. Our experiments were conducted on a cluster of 18 nodes, each equipped with 2 AMD EPYC 7543 32-core CPUs with 256 MB L3 cache, with 1TB of memory per node, and running on a Rocky Linux 9.3 operating system. We used HEBO version 0.3.5, YAHPO Gym version 1.0.1 and JAHS-Bench-201 version 1.1.0. Our experiments required approximately 100 000 CPU hours.

# 5 RESULTS AND DISCUSSION

In this section, we present and discuss the experimental results comparing our dynamic ensembling approach to the baselines. We first present the rank results on different benchmarks. We then analyse the evolution of the weights assigned to surrogate models throughout the optimisation process, and finally assess the running time of our method.

## 5.1 OVERALL PERFORMANCE

We present the average rank as a function of the percentage of the budget for YAHPO Gym in Figure 2. The results are split into four parts according to the mean time required for evaluating the target function. We thus show mean ranks on all target functions, on cheap functions with a budget of up to 10 minutes, on medium-cost functions with a budget between 10 minutes up to 1 hour, and on expensive functions with a budget exceeding 1 hour. An analysis of the statistical significance of our results is presented in Appendix I. When considering all target functions, we see that all dynamic ensembling approaches outperform the baselines, both single-surrogate ones and the static

ensemble with equal weights. Using the weighting scheme with $\alpha = 1.0$ is top-ranked on average, followed by $\alpha = 0.9$.

When it comes to cheap target functions, GP is the best-performing surrogate for very low budgets (*i.e.*, less than $60\%$ of the 10-minute budget). As the budget increases, our dynamic ensembling approach with $\alpha = 1.0$ outperforms every surrogate on average. Ensembling approaches with smaller $\alpha$ values perform worse; ensembles with $\alpha = 0.9$ and $\alpha = 0.5$ outrank GP only in the last $10\%$ of the 10-minute budget. This highlights the fact that dynamically pruning of surrogates with null weights within the ensemble enhances performance for low budgets, as the running time required for optimisation is reduced. For medium-cost and expensive functions, all dynamic ensembling approaches exhibit better ranks on average. We observe that the ensemble using $\alpha = 1.0$ values is now the worst performer for these functions, giving way to the other values of $\alpha$. Moreover, we see that dynamic ensembling outranks a static ensemble with equal weights.

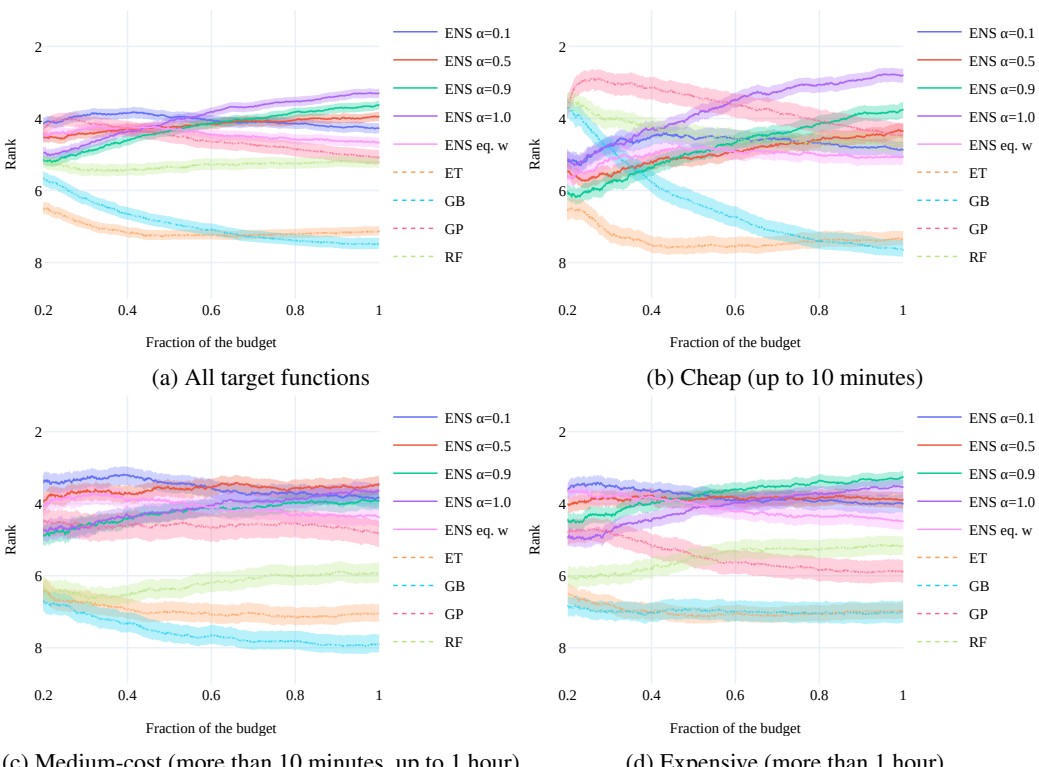

(a) All target functions

(b) Cheap (up to 10 minutes)

(c) Medium-cost (more than 10 minutes, up to 1 hour)

(d) Expensive (more than 1 hour)

Figure 2: Mean ranks of HEBO with different surrogate models on YAHPO Gym and JAHS-Bench-201, split according to different budgets: (a) all target functions, (b) cheap target functions with a budget of up to 10 minutes, (c) medium-cost target functions with a budget between 10 minutes and up to 1 hour, (d) expensive target functions with a budget of more than 1 hour. Ensembling methods are displayed with solid lines. Single surrogates are displayed with dashed lines. Ensembling methods outrank single surrogates in all budgets. Which $\alpha$ value in the ensemble is top-ranked depends on the budget. On $y$-axis: 1 is the best possible rank value.

## 5.2 WEIGHTS EVOLUTION

We present the evolution of the weights assigned to surrogate models within the ensemble on NAS-Bench-301 from YAHPO Gym in one run using 256 target function evaluations in Figure 3. In order to obtain consistent visualisation of the weight values, here, we switch to measuring the optimisation budget in terms of function evaluations rather than running time. Full results in the setting with the total budget of 256 evaluations can be found in Appendix F. In Figure 3, We observe that higher values of $\alpha$ yield more rapid and frequent changes of weights. We also notice that in different stages of the optimisation, different surrogate models are given precedence. For example, for $\alpha = 1.0$, RF is used in the beginning of the optimisation (for a low number of evaluations), while GB is favoured

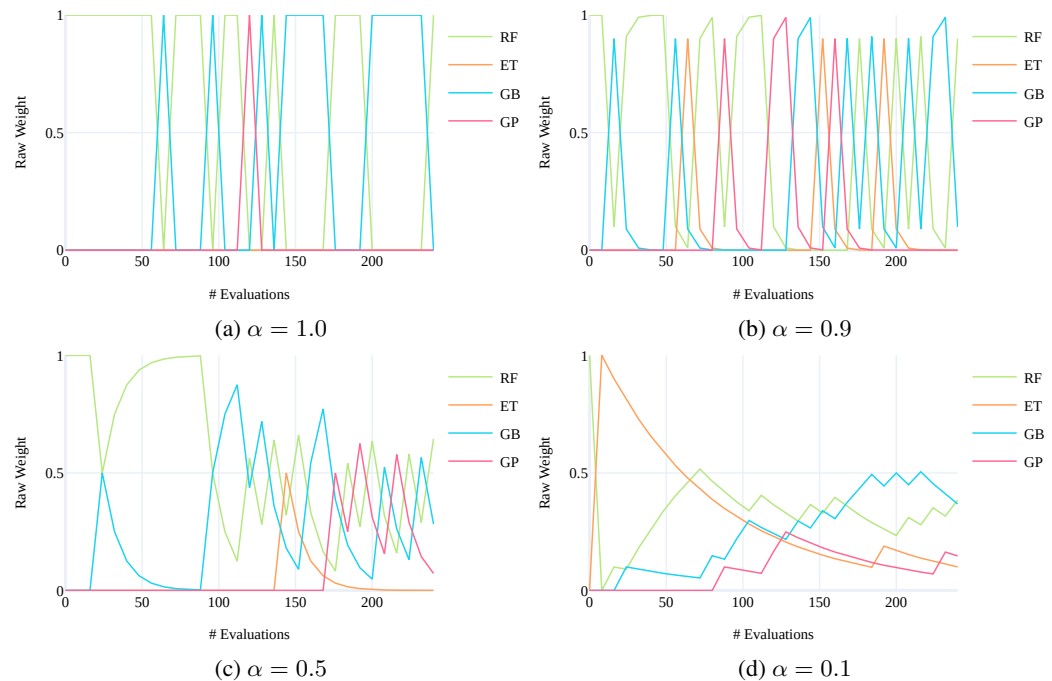

(a) $\alpha = 1.0$

(b) $\alpha = 0.9$

(c) $\alpha = 0.5$

(d) $\alpha = 0.1$

Figure 3: Raw weights of the ensembles with different $\alpha$ values on NAS-Bench-301 (part of YAHPO Gym). A higher value of $\alpha$ causes sharper variations in the weights. In different stages of the optimisation, a different surrogate model gets a higher weight.

more often towards the end of the optimisation (for a higher number of evaluations). This shows that the dynamic weight assignment adapts the weighting scheme throughout the optimisation process.

The increased adoption of GB in the later stages of the optimisation process is an interesting observation, even though GB as a standalone surrogate consistently ranks as the least effective model. One potential explanation is that GB may lead to inaccurate predictions when working with a small number of samples. However, as more samples are collected, particularly when GB gets refined with more observed points around the optimum, the accuracy of its predictions improves.

## 5.3 RUNNING TIME

We examine the wall-clock time used by the optimiser itself (i.e., excluding the time required to evaluate the target function) as a percentage of the optimisation budget and as absolute running time relative to the number of target function evaluations. Figure 4a shows the proportion of the optimisation budget (in seconds) used by the optimiser to suggest the next configurations to sample across all functions (from both YAHPO Gym and JAHS-Bench-201). As expected, for very low budgets (up to approximately 20 seconds), the optimiser consumes a high fraction of the budget (more than 35%). As the budget increases, this fraction decreases, until it becomes negligible (less than 1% for budgets of more than 10 000 seconds). We observe that using ensembles requires a higher fraction of the optimisation budget than using a single surrogate. However, the difference becomes indistinguishable when the budget exceeds 10 000 seconds. When using dynamic ensembling with $\alpha = 1.0$, the difference becomes difficult to discern even with a budget of 1 000 seconds, once more demonstrating the effectiveness of dynamic pruning of surrogate models.

Additionally, we assess the absolute cost of using different surrogates in Figure 4b. Ensembling approaches take the longest to run, and among them using $\alpha = 1.0$ requires the least time due to dynamic pruning. We also notice that ensembling approaches take roughly twice as long as tree-based surrogates. GPs are the fastest to run with the low number of evaluations, however the gap between them and tree-based methods shrinks substantially as more observations are added.

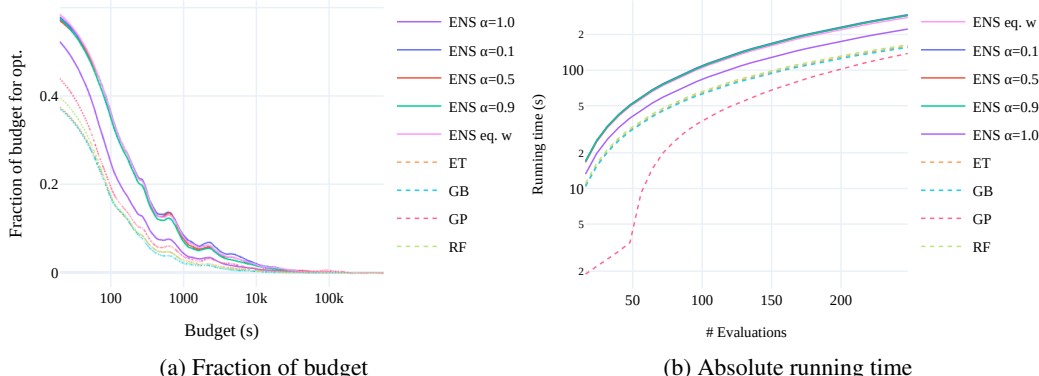

(a) Fraction of budget        (b) Absolute running time

Figure 4: Running time of the optimisation with dynamic ensembling methods (excluding the time needed to evaluate the target function): (a) as a fraction of optimisation budget, and (b) as absolute running time relative to the number of target function evaluations. Ensembles generally require a higher fraction of the budget compared to single surrogates and take the longest to run in absolute terms, with a notable exception of the ensemble with $\alpha = 1.0$, which consumes less budget by an order of magnitude, and which is the fastest to run, due to dynamic pruning.

## 6   CONCLUSIONS, LIMITATIONS AND FUTURE WORK

In this work, we proposed a novel dynamic ensembling approach for surrogate models in hyperparameter optimisation. Our method constructs a weighted ensemble of different types of surrogate models by assessing the accuracy of different surrogates throughout the optimisation process and dynamically determining which weight should be assigned to each surrogate model. We experimented with a wide variety of HPO benchmark tasks and found that our method results in a better mean rank than single-surrogate baselines and a static ensemble with equal weights.

Despite very promising results of our work, there is still room for improvement. One limitation is the higher running time required to both train and evaluate the surrogate models and to optimise the acquisition function compared to a single-surrogate BO. We mitigate this by introducing dynamic pruning of surrogates, which substantially reduces the required running time. However, dynamic pruning only occurs when we have models with null weights, which is rare in ensembles with values of $\alpha$ smaller than $1.0$. Furthermore, we observe that the Gaussian process outperforms our method for very small budgets (less than 5 minutes). This is in line with the fact that BO equipped with GP excels precisely in a very low-budget setting, as well as that our method requires higher time for the training additional surrogate models.

Our method itself comes with a hyperparameter, the smoothing factor $\alpha$. While different values of $\alpha$ work best with different target functions, we observed that using $\alpha = 1.0$ works consistently well on all functions, regardless of the cost of function evaluations. The impact of historical accuracy measurements thus seem to contribute only marginally when determining the weights.

Our work presented here opens several avenues for future research. Multi-fidelity approaches are very commonly used for HPO, especially in expensive tasks where each evaluation on full fidelity can take more than an hour. It seems promising to extend our method to multi-fidelity scenarios. A further promising direction is to learn in which scenarios (*e.g.,*, input dimensionality, phase of the optimisation) which surrogate model(s) work best and to create an ensemble by training only these specific models; in case a single surrogate model is chosen, the overhead of ensembling would thus be eliminated.

Overall, we believe that combining multiple surrogate models in BO is a promising direction for achieving better surrogate predictions, thus allowing for faster convergence and increased sample efficiency. `DensBO` is the first method that dynamically leverages the potential of complementary surrogates within the BO procedure itself.

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

# A  SOCIETAL IMPACT

This paper presents fundamental empirical work whose goal is to advance the field of hyperparameter optimisation, and (automated) machine learning more broadly. We do not see any negative ethical and societal implications of our work. A positive impact of our work concerns reducing computational load for hyperparameter optimisation, as it is able to find well-performing configurations in less time than standard baselines we compare against. This leads to a reduced carbon footprint and saved energy resources.

# B  CODE AVAILABILITY AND REPRODUCIBILITY

Our code, as well as full results on all benchmarks and additional figures, can be found at the anonymous Git repository: `https://anonymous.4open.science/r/dyn_ens_supp-D5C2/`.

# C  DETAILED IMPLEMENTATION DESCRIPTION

Our method is implemented in HEBO (Cowen-Rivers et al., 2022), a state-of-the-art BO-based optimiser (Eggensperger et al., 2021). HEBO starts by sampling random configurations based on Sobol sequences. Then, in each BO iteration, it transforms the (input) configurations and the (output) performances to tackle non-stationarity and heteroscedasticity of the data, respectively. Non-stationarity of the input means that the GP kernel does not only depend on the norm between two inputs; it is corrected by appropriate input warping, in this case the Kumaraswamy transformation. Given the dimensionality of the decision variable $d$, tuneable warping parameters for each dimension $a_k$ and $b_k$, and a vector concatenating all free parameters $\gamma$, the Kumaraswamy warping is defined as follows for all input dimensions:

$$\left[\text{Kumaraswamy}_\gamma\left(\boldsymbol{x}_l\right)\right]_k = 1 - (1 - [\boldsymbol{x}_l]_k^{a_k})^{b_k} \ \forall k \in [1:d] \tag{9}$$

Heteroscedasticity of the output means that the output does not adhere to a Gaussian noise model, but that the noise is a function of the input, *i.e.*, depending on the input, the noise is prone to changing around the mean. The performances are thus transformed using power transformations, either Box-Cox (which supports either strictly positive or strictly negative inputs) or Yeo-Johnson (which handles arbitrary inputs). Given a tunable transformation parameter $\zeta$, the Box-Cox transformation applies the following mapping to each of the outputs:

$$\text{B.C.}_\zeta\left(y_l\right) = y_l^\zeta - 1/\zeta \ \text{for} \ \zeta \neq 0 \ \text{and} \ \text{B.C.}_\zeta\left(y_l\right) = \log y_l \ \text{if} \ \zeta = 0 \tag{10}$$

where $y_l$ is the performance of the $l^{th}$ hyperparameter configuration. The Yeo-Johnson transformation is defined as follows:

$$\text{Y.J.}\zeta\left(y_l\right) = \begin{cases} \frac{(y_l+1)^\zeta - 1}{\zeta}, & \text{if } \zeta \neq 0, y_l \geq 0 \\ \log\left(y_l + 1\right), & \text{if } \zeta = 0, y_l \geq 0 \\ \frac{(1-y_l)^{2-\zeta}-1}{\zeta-2} & \text{if } \zeta \neq 2, y_l < 0 \\ -\log\left(1-y_l\right) & \text{if } \zeta = 2, y_l < 0 \end{cases} \tag{11}$$

Power transformations are used to give the data a zero mean and a variance of 1. This is done in order to get the data distribution close to a Gaussian distribution, which improves the fit of GPs to the data. HEBO then fits the GP and Kumaraswamy-transformed parameters using the well-known Adam optimiser.

After fitting the surrogate model, HEBO optimises a multi-objective acquisition function consisting of three widely used acquisition functions: EI, PI and UCB, which are defined as follows:

$$\text{EI: } \alpha_{\text{EI}}^{\boldsymbol{\theta}}\left(\boldsymbol{x}_{1:q} \mid \mathcal{D}\right) = \mathbb{E}_{\text{posterior}}\left[\max_{j \in 1:q}\left\{\text{ReLU}\left(f\left(\boldsymbol{x}_j\right) - f\left(\boldsymbol{x}^+\right)\right)\right\}\right] \tag{12}$$

$$\text{PI: } \alpha_{\text{PI}}^{\boldsymbol{\theta}}\left(\boldsymbol{x}_{1:q} \mid \mathcal{D}\right) = \mathbb{E}_{\text{posterior}}\left[\max_{j \in 1:q}\left\{\mathbb{1}\left\{f\left(\boldsymbol{x}_j\right) - f\left(\boldsymbol{x}^+\right)\right\}\right\}\right] \tag{13}$$

$$\text{UCB: } \alpha_{\text{UCB}}^{\boldsymbol{\theta}}\left(\boldsymbol{x}_j\right) = \mathbb{E}_{\text{posterior}}\left[\max_{j \in 1:q}\left\{\mu_{\boldsymbol{\theta}}\left(\boldsymbol{x}_j\right) + \sqrt{\beta\pi/2}\left|\gamma_{\boldsymbol{\theta}}\left(\boldsymbol{x}_j\right)\right|\right\}\right] \qquad (14)$$

where $\boldsymbol{x}_j$ is the $j^{th}$ vector of $\boldsymbol{x}_{1:q}$; $\boldsymbol{x}^+$ is the best performing input in the data so far; $\mathbb{1}\!\!\!1\{\cdot\}$ is the left-continuous Heaviside step function; $\mu_{\boldsymbol{\theta}}\left(\boldsymbol{x}_j\right)$ is the posterior mean of the predictive distribution; and $\gamma_{\boldsymbol{\theta}}\left(\boldsymbol{x}_j\right) = f\left(\boldsymbol{x}_j\right) - \mu_{\boldsymbol{\theta}}\left(\boldsymbol{x}_j\right)$. HEBO searches for candidates that are in the Pareto front of the three functions. This leverages the fact that for different functions and stages of the optimisation process, different acquisition functions work better (Benjamins et al., 2022a). HEBO then finds candidate configurations using a multi-objective optimiser NSGA-II.

We use HEBO with a batch size of 8 (*i.e.*, retraining the surrogate every 8 epochs), as suggested in the HEBO documentation[1], and as done in other hyperparameter optimisers such as SMAC (Lindauer et al., 2019). For the Gaussian process, we use the default implementation and hyperparameters of HEBO. For tree-based models, we use the default hyperparameters of `scikit-learn`, with the exception of the number of trees which we set to 10, as done in SMAC. In some cases, GPs return an error during fit. If this occurs when using ensembling, at iteration $t$, we assign:

$$w_{t,GP} = 0 \qquad (15)$$

Otherwise, when using GPs alone, we instead use random search rather than BO to decide on the next configurations to sample.

## C.1 USED SOFTWARE

In our implementation, we used the following packages:

- HEBO: `https://github.com/huawei-noah/HEBO/tree/master/HEBO` version 0.3.5 under MIT license.

- SCIKIT-LEARN: `https://scikit-learn.org/` version 1.4.1 under BSD-3 license.

- YAHPO Gym: `https://github.com/slds-lmu/yahpo_gym` version 1.0.1 under Apache 2.0 license.

- JAHS-Bench-201: `https://github.com/automl/jahs_bench_201/tree/main` version 1.1.0 under MIT license.

## C.2 MOTIVATION FOR USING EXPONENTIAL MOVING AVERAGE

We use an exponential moving average inspired by the reinforcement learning algorithm Q-learning (Watkins, 1989), which uses an exponential moving average to update its policy in every iteration. In Q-learning, the $Q(S_t, A_t)$ of a state $S_t$ and an action $A_t$ holds the expected reward of the agent by being in state $S_t$ and taking action $S_t$. Q-learning then proceeds iteratively by exploring the state space and updating the Q-values according to the observed rewards. By analogy, in our optimisation case, we attempt to learn a "weighting policy" for the weights in our models. However, a notable difference between the reinforcement learning case and the dynamic ensembling is that reinforcement learning learns a single, static, policy in a training phase. We learn the policy online, during the optimisation process.

Another motivation for the usage of the exponential moving average is to focus on the accuracy of the surrogate models in the region that is currently favoured by the acquisition function. This region can change during the optimisation process. With the exponential moving average, intuitively speaking, we can gradually "forget" about old scores and put a higher emphasis on new ones.

## C.3 SINGLE SURROGATE MODEL BASELINES

In Section 2.2, we gave an introduction on GPs. In this appendix, we elaborate on the three other surrogate models we used in the paper.

Random forest (RF) (Breiman, 2001) is a *bagged* ensemble of decision trees. This means that each tree is fitted on a different random subset of the data. This then helps avoid overfitting and improve

---

[1]`https://hebo.readthedocs.io/en/latest/optimisation.html`

the prediction accuracy. Random forests are known as good performance predictors (Hutter et al., 2014) and have been used in BO pipelines as surrogate models (Hutter et al., 2011).

Extremely randomised trees (Geurts et al., 2006), also known as extra trees (ET,) are an ensemble of random trees, where the splitting threshold is determined randomly, instead of based on a predetermined criterion. Furthermore, each tree is built on the whole dataset. Extra trees are known to have lower variance than random forests.

Gradient boosting (GB) (Friedman, 2001) is an ensembling method that consists of multiple decision trees. Gradient boosting iteratively constructs decision trees to minimise the loss of the ensemble.

## D  BENCHMARK DETAILS

Yet Another HPO Benchmark (YAHPO) Gym (Pfisterer et al., 2022) is a surrogate-based benchmark for hyperparameter optimisation on tabular machine learning tasks. The benchmark consists of multiple datasets from OpenML and various machine learning algorithms. The authors provide a surrogate model for each machine learning algorithm and a dataset that predicts different objectives, such as accuracy, training time, and more.

YAHPO Gym contains several scenarios. Each scenario has a configuration space and is based on a machine learning algorithm. Each scenario contains multiple datasets, which are called instances. The YAHPO Gym scenarios are:

- rbv2_ranger (Binder et al., 2020): random forest using the ranger R implementation.
- rbv2_rpart (Binder et al., 2020): decision tree using the mlr implementation.
- rbv2_glmnet (Binder et al., 2020): generalised linear models (GLM) with elastic net regularisation using the mlr implementation.
- rbv2_svm (Binder et al., 2020): support vector machine (SVM) with the mlr implementation.
- rbv2_xgboost (Binder et al., 2020): gradient boosting using XGBoost.
- rbv2_aknn (Binder et al., 2020): k-nearest neighbours using the mlr implementation.
- rbv2_super (Binder et al., 2020): a combined algorithm selection and hyperparameter optimisation (CASH) scenario that uses all the above rbv2 scenarios.
- iaml_glment (Pfisterer et al., 2022): generalised linear models (GLM) with elastic net regularisation using the mlr implementation.
- iaml_rpart (Pfisterer et al., 2022): decision tree using the mlr implementation.
- iaml_ranger (Pfisterer et al., 2022): random forest using the ranger R implementation.
- iaml_xgboost (Pfisterer et al., 2022): gradient boosting using XGBoost.
- iaml_super (Pfisterer et al., 2022): combined algorithm selection and hyperparameter optimisation (CASH) scenario that uses all the above iaml scenarios.
- LCBench (Zimmer et al., 2021): optimising the training hyperparameters and architecture of multi-layer perceptron on tabular datasets.
- FCNet (Falkner et al., 2018): hyperparameter optimisation of a fully-connected network for tabular data.
- NAS-Bench-301 (Zela et al., 2022): neural architecture search (NAS) scenario using a constant set of training hyperparameters. The search space describes a cell of the neural network and the hyperparameters are different operations done in each cell.

The configuration spaces for the rbv2 scenarios are in Table 1. For rbv2_super, the configuration space contains all the individual algorithms configuration spaces, with an additional hyperparameter which selects the algorithm. The configuration spaces for the iaml scenarios are in Table 5, the iaml_super scenario is similar to rbv2_super, where there is an additional hyperparameter to select to algorithm. While the iaml and rbv2 scenarios look similar, there are a few differences in the configuration space. First, the configuration spaces of rbv2 contains the input imputation used, while in iaml the input imputation is constant. Another difference is a different ranges of hyperparameters

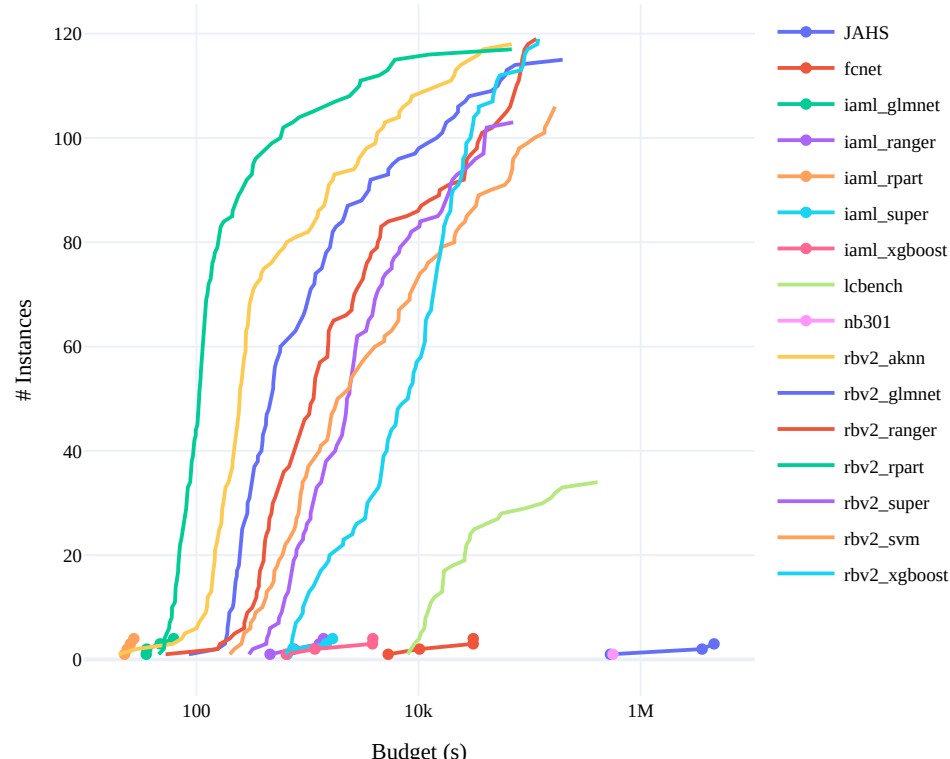

Figure 5: Cumulative number of instances as a function of the budget for every instance, for each scenario. *E.g.*, 60 instances from the iaml_glmnet scenario (in dark green) take up to 100 seconds to evaluate.

(for example, the number of rounds in xgboost). In all scenarios, the target metric is accuracy, as defined in (Pfisterer et al., 2022). The search spaces for LCBench, FCNet and NAS-Bench-301 are shown in Table 3, Table 4 and Table 2, respectively.

### D.1 EVALUATION BUDGET

All YAHPO Gym and JAHS-Bench-201 surrogates predict the running time required to train and evaluate the model using a specific configuration. Therefore, we calculate the budget per instance similarly to Eggensperger et al. (2021): we sample 1 000 random configurations per instance and predict the required running time. We then calculate the mean running time per instance. The budget for each optimisation is then $100\times$ mean running time of the instance. The running times per scenario are available in Figure 5.

## E    FULL RESULTS

All figures (convergence plots, weight evolution and performance comparison), including results for JAHS-Bench-201, are available in the supplemental Git repository (see Appendix B) due to reasons of space, as there are 859 such figures. We provide convergence plots in Figure 6 for a subset of YAHPO Gym, namely the YAHPO-SO benchmark suite that contains 20 HPO problems from different scenarios.

## F    RESULTS WITH 256 EVALUATIONS

In addition to the main results in Section 5.1, we performed additional experiments to assess the effectiveness of our method as a function of the number of evaluations and not as the budget measured in wall-clock time. We used 8 initial samples and a total budget of 256 evaluations. This

Table 1: Configuration spaces for the rbv2 scenarios of YAHPO Gym.

| Algorithm | Hyperparameter | Range | Comments |
|---|---|---|---|
| rbv2_glment | alpha | [0, 1] | |
| | s | [0.001, 1097] | log-scaled |
| | imputation | {mean, median, hist} | |
| rbv2_rpart | cp | [0.001, 1] | log-scaled |
| | maxdepth | [1, 30] | |
| | minbucket | [1, 100] | |
| | minsplit | [1, 100] | |
| | imputation | {mean, median, hist} | |
| rbv2_svm | kernel | {linear, polynomial, radial} | |
| | cost | [4.5e-0.5, 2.2e4] | log-scaled |
| | gamma | [4.5e-05, 2.2e4] | log-scaled conditional by the kernel |
| | tolerance | [4.5e-0.5, 2] | log-scaled |
| | degree | [2, 5] | conditional by the kernel |
| | imputation | {mean, median, hist} | |
| rbv2_aknn | k | [1, 50] | |
| | distance | {l2, cosine, ip} | |
| | M | [18, 50] | |
| | ef | [7, 403] | log-scaled |
| | ef_construction | [7, 403] | log-scaled |
| | imputation | {mean, median, hist} | |
| rbv2_ranger | num. trees | [1, 2000] | |
| | sample fraction | [0.1, 1] | |
| | mtry power | [0, 1] | |
| | respect unordered factors | {ignore, order, partition} | |
| | min node size | [1, 100] | |
| | splitrule | {gini, extratrees} | |
| | num random splits | [1, 100] | |
| | imputation | {mean, median, hist} | |
| rbv2_xgboost | booster | {gblinear, gbtree,dart} | |
| | nrounds | [7, 2980] | |
| | eta | [0.001, 1] | |
| | gamma | [4.5e-05, 7.4] | |
| | lambda | [0.001, 1097] | |
| | max_depth | [1, 15] | |
| | min_child_weight | [2.72, 148.4] | |
| | colsample_bytree | [0.01, 1] | |
| | rate_drop | [0, 1] | |
| | skip_drop | [0, 1] | |
| | imputation | {mean, median, hist} | |

Table 2: Configuration spaces for the NAS scenarios: the NAS-Bench-301 scenario of YAHPO Gym, and JAHS-Bench-201.

| Benchmark | Hyperparameter | Range | Comments |
|---|---|---|---|
| JAHS-Bench-201 | Op[1-6] | {skip-connect, zero, 1x1 conv, 3x3 conv 3x3 avg pool} | |
| | Activation | {ReLU, Hardswish, Mish} | |
| | Learning Rate | [10^-3, 10^0] | log-scaled |
| | Weight Decay | [10^-5, 10^-2] | log-scaled |
| | Trivial Augment | {On, Off} | |
| NAS-Bench-301 | edge_normal_{0-13} | {max pool 3x3, avg pool 3x3, skip connect, sep conv 3x3, sep conv 5x5, dil conv 3x, dil conv 5x5} | |
| | edge_reduce_{0-13} | {max pool 3x3, avg pool 3x3, skip connect, sep conv 3x3, sep conv 5x5, dil conv 3x, dil conv 5x5} | |
| | COLON_inputs_node_normal_{3-5} | {0_1, 0_2, 1_2} | |
| | COLON_inputs_node_reduce_{3-5} | {0_1, 0_2 1_2} | |

Table 3: Configuration spaces for the LCBench scenarios of YAHPO Gym.

| Benchmark | Hyperparameter | Range | Comments |
|---|---|---|---|
| LCBench | batch size | [16, 512] | log-sacled |
| | learning rate | [1e-4, 0.1] | log-sacled |
| | momentum | [0.1, 0.9] | |
| | weight decay | [1e-5, 0.1] | |
| | num layers | [1, 5] | |
| | max_units | [64, 1024] | log-sacled |
| | max_dropout | [0, 1] | |

Table 4: Configuration spaces for the FCNet scenarios of YAHPO Gym.

| Benchmark | Hyperparameter | Range | Comments |
|---|---|---|---|
| FCNet | activation_fn_1 | [tanh, relu] | |
| | activation_fn_2 | [tanh, relu] | |
| | batch_size | [8, 64] | log-scaled |
| | dropout_1 | [0.0, 0.6] | |
| | dropout_2 | [0.0, 0.6] | |
| | epoch | [1, 100] | log-sacled |
| | init_lr | [0.0005, 0.1] | log-scaled |
| | lr_schedule | const, cosine | |
| | n_units_1 | [16, 512] | |
| | n_units_2 | [16, 512] | log-scaled |
| | replication | [1, 4] | |

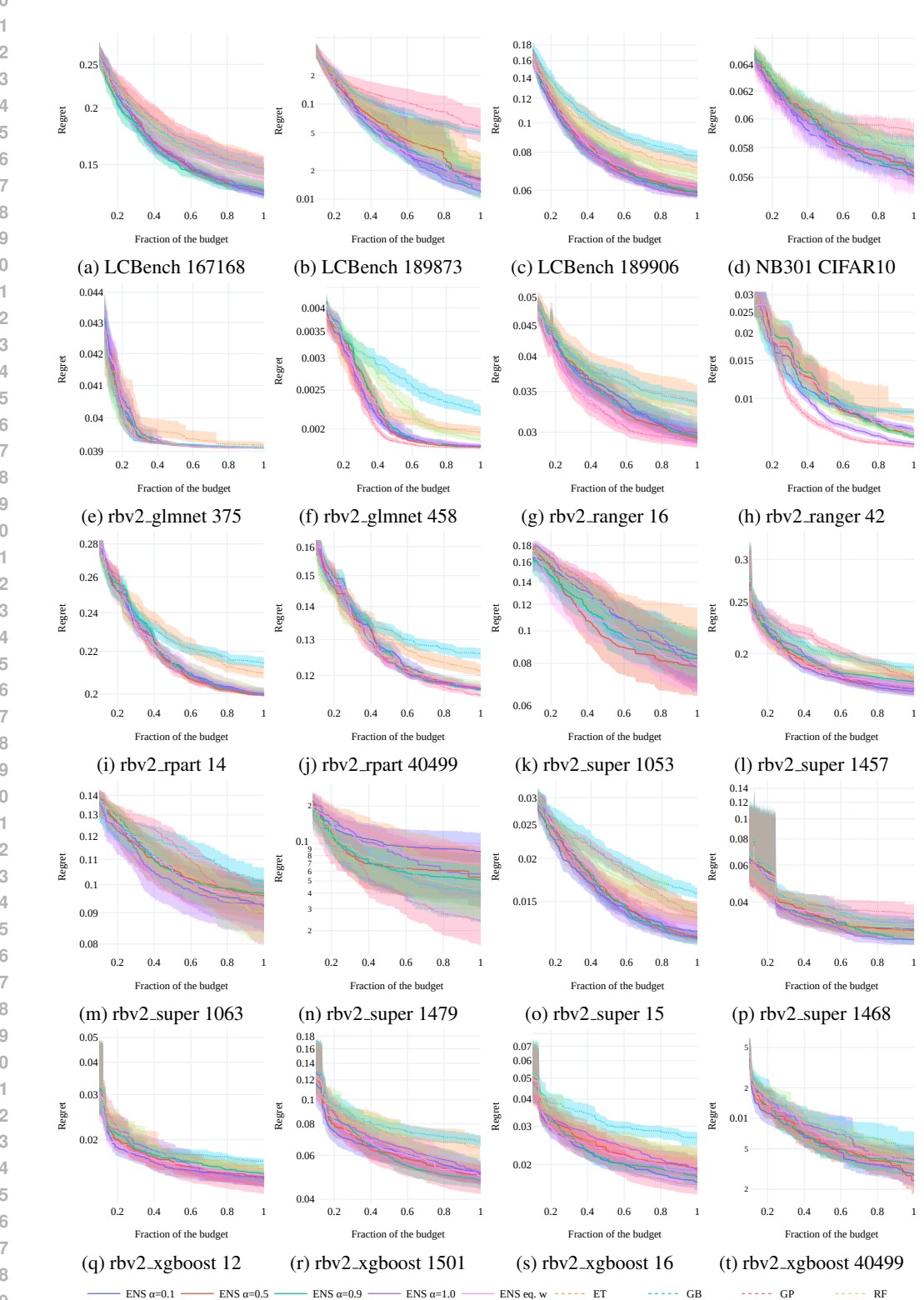

Figure 6: Convergence curves (regret over time) of dynamic ensembling approaches and various baselines on the YAHPO-SO benchmark set. In each sub-figure caption, the first part is the scenario and the second part is the instance (*i.e.*, dataset id). On $y$-axis: lower regret is better.

Table 5: Configuration spaces for the iaml scenarios of YAHPO Gym.

| Benchmark | Hyperparameter | Range | Comments |
|---|---|---|---|
| iaml_glmnet | alpha | [0, 1] | |
| | s | [1e-4, 1000] | log-scaled |
| | cp | [1e-4. 1] | log-scaled |
| iamb_rpart | maxdepth | [1, 30] | |
| | minbucket | [1, 100] | |
| | minsplit | [1, 100] | |
| | num trees | [1, 2000] | |
| iaml_ranger | replace | {True, False} | |
| | sample fraction | [0., 1] | |
| | mtry ratio | [0, 1] | |
| | respect unordered factors | {ignore, order, partition} | |
| | min node size | [1, 100] | |
| | splitrule | {gini, extratrees} | |
| | num random splits | [1, 100] | |

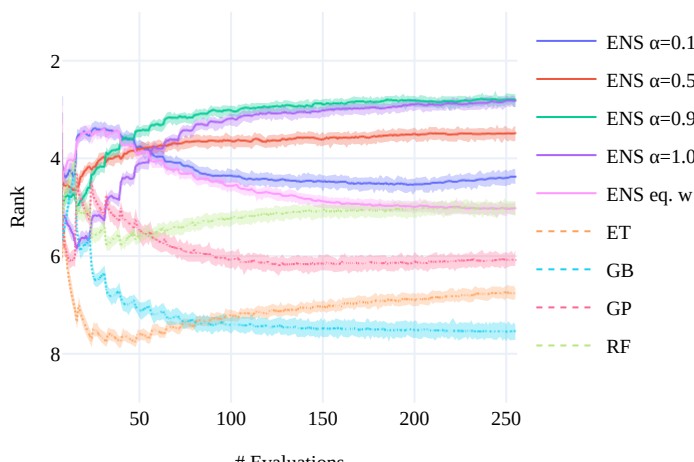

Figure 7: Mean ranks of HEBO with different surrogate models when using 256 evaluations on YAHPO Gym and JAHS-Bench-201. Dynamic ensembling approaches dominate the other surrogate models.

additional experiment was designed to show the optimiser's ability to find good performing candidates, regardless of the running time required for the optimiser or the target function. We evaluated the approach on both YAHPO Gym and JAHS-Bench-201. The results are presented in Figure 7. Dynamic ensembling is ranked better, with $\alpha = 0.9$ having the best performance, closely followed by $\alpha = 1.0$. All dynamic ensembling methods are ranked better than the single surrogate baselines, as well as the static ensemble with equal weights to all models.

## G   RESULTS WITH 1024 EVALUATIONS

We provide results for HEBO with different surrogate models on a (relatively) high-budget setting of 1024 total evaluations and an initial design size of $2d$, where $d$ is the number of hyperparameters. The results are shown in Figure 8. Similarly to the previously shown results, dynamic ensembling outranks the baselines, including the static ensemble baseline. Using $\alpha = 0.9$ achieves the best average rank, with a larger gap from $\alpha = 1.0$ than in the 256 evaluations setting. These results show that our dynamic ensembling method works well also in a high-budget setting.

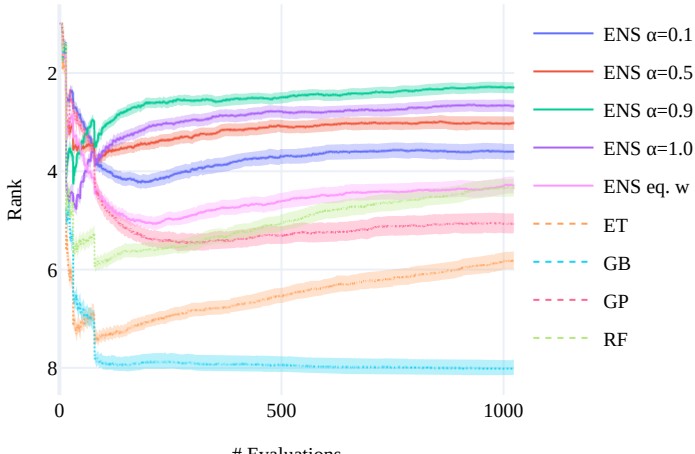

Figure 8: Mean ranks of HEBO with different surrogate models when using 1024 evaluations on YAHPO Gym and JAHS-Bench-201. Dynamic ensembling approaches achieve better rank than single surrogate models and static ensemble.

## H USING DIFFERENT WEIGHTS FOR VARIANCE

We experimented with weighting the mean and the variance separately by having two, independent weight vectors. The weight vector of the mean is the same as described in Section 3. For the variance weights, we define the variance error VE for a model $m \in M$ as follows:

$$\text{VE}_m(\lambda_t) = \frac{1}{k} \sum_{i=1}^{k} \min\{|(\mu_m(\lambda_t^{(i)}) - \sigma_m(\lambda_t^{(i)})) - c(\lambda_t^{(i)})|, |(\mu_m(\lambda_t^{(i)}) + \sigma_m(\lambda_t^{(i)})) - c(\lambda_t^{(i)})|\} \quad (16)$$

Intuitively, it is the distance to the closest variance bound, as can be seen in Figure 9. We update the weights for variance accordingly. The new weights are defined as:

$$\bar{w}'_{t,m} = \begin{cases} 1 & \text{if } \text{VE}_m(\lambda_t) = \min_{l \in M} \text{VE}_l(\lambda_t), \\ 0 & \text{otherwise,} \end{cases} \quad (17)$$

We calculate the weights for variances using the exponential moving average:

$$w'_{t+1,m} = (1 - \alpha) \cdot w'_{t,m} + \alpha \cdot \bar{w}'_{t,m}, \quad (18)$$

Then, we normalise the weights for variances independently from the weights for the means:

$$\hat{w}'_{t,m} = \frac{w'_{t,m}}{\sum_{j \in M} w'_{t,j}}, \quad (19)$$

Finally, the variance predicted by the ensemble is defined as the weighted sum of the normalised weights of the variances:

$$\sigma_{\text{ens}}(\lambda) = \sum_{m \in M} \hat{w}'_{t,m} \cdot \sigma_m(\lambda). \quad (20)$$

We run the dynamic ensembling approach with variance with $\alpha = 0.9$ and a budget of $100\times$ mean evaluation time per target function. Similarly to the results presented in Section 5.1, we present the mean rank of all methods, including dynamic ensembling with different weights for variances in Figure 9. The experimental setup is similar to Section 5.1, where the budget is $100\times$ mean running time of one evaluation. We see that using different weights for variance ranks worse than the similar dynamic ensembling with the same weights for both the means and variances.

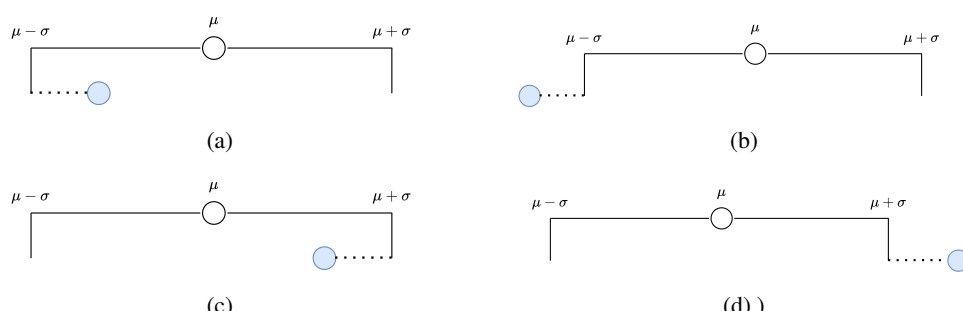

Figure 9: Illustration of the calculation of the variance error. The white circle is the the predicted mean, and the blue dot is the actual value of the target function.

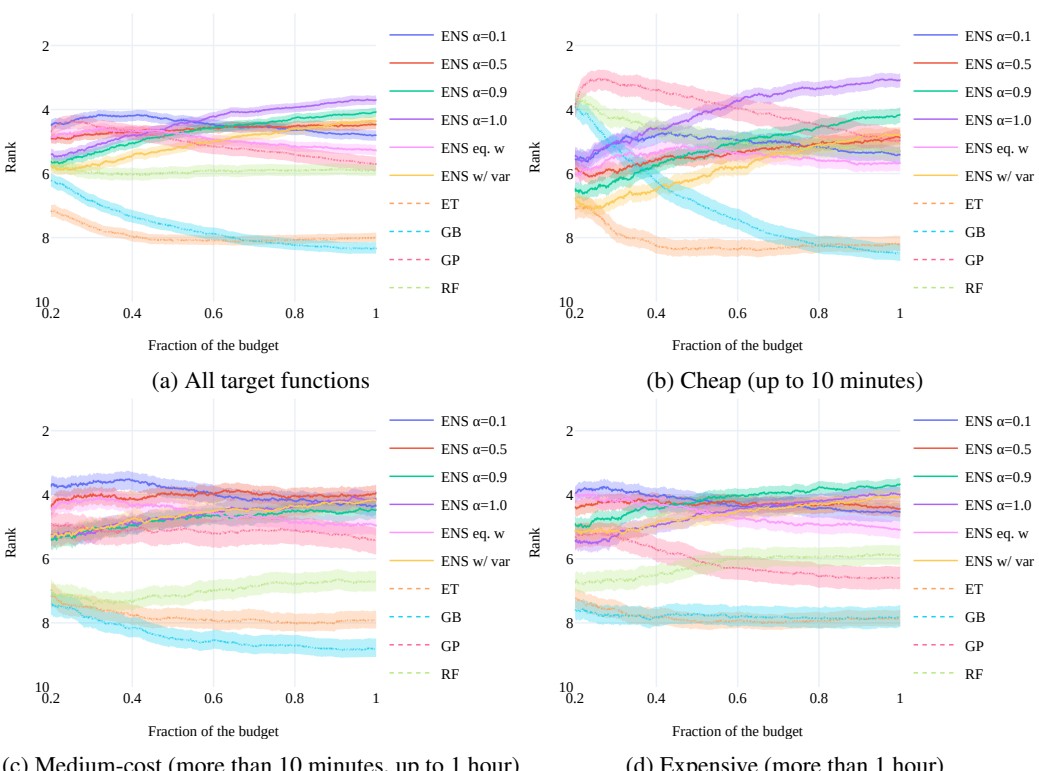

Figure 10: Mean ranks of HEBO with different surrogate models on YAHPO Gym and JAHS-Bench-201 (as in Figure 2) *with an additional baseline: a dynamic ensemble with different weighting schemes for mean and variance (in yellow)*, split according to different budgets: (a) all target functions, (b) cheap target functions with a budget of up to 10 minutes, (c) medium-cost target functions with a budget between 10 minutes and up to 1 hour, (d) expensive target functions with a budget of more than 1 hour. The ensemble with different weights for mean and variance closely follows the trend of other dynamic ensembling approaches, but does not outrank them.

## I  STATISTICAL SIGNIFICANCE

We test the statistical significance of our results presented in Section 5.1, and show that dynamic ensembling significantly outperforms single-surrogate-based BO. To do this, we use the best-found target function values per optimiser. We then calculate the mean value for each function and optimiser pair over the 51 random seeds. The optimiser that has the best value per function is then the best for the function. We calculate whether the values obtained for each of the other optimisers are statistically equal to the values of the best optimiser using a permutation test with $10\,000$ samples and a significance level of $0.05$. We report these results in Table 6. We see that dynamic ensembling approaches are equal to the best method more times than any other baseline, across all budget groups. As expected, on cheap functions, $\alpha = 1.0$ equals the most times to the best method. For medium-cost functions, $\alpha = 0.5$ performs best, while for the expensive ones, $\alpha = 0.9$ is the best. We can, therefore, conclude that using dynamic ensembling is significantly better than any other baseline.

Table 6: Critical differences of different surrogate models. Total is the number of instances in each category.

| Model | All | Cheap | Medium | Expensive |
|---|---|---|---|---|
| ENS $\alpha = 0.1$ | 587 | 152 | 194 | 241 |
| ENS $\alpha = 0.5$ | 628 | 177 | **201** | 250 |
| ENS $\alpha = 0.9$ | 657 | 197 | 192 | **268** |
| ENS $\alpha = 1.0$ | **687** | **247** | 195 | 245 |
| ENS eq. w. | 519 | 144 | 172 | 203 |
| ET | 198 | 53 | 67 | 78 |
| GB | 164 | 44 | 33 | 87 |
| GP | 492 | 201 | 147 | 144 |
| RF | 396 | 145 | 89 | 162 |
| Total | 859 | 312 | 234 | 313 |

