# OpenReview forum: "DensBO: Dynamic Ensembling of Surrogate Models for Hyperparameter Optimisation"
_ICLR.cc/2025/Conference — Submitted to ICLR 2025_

### Official Review · Reviewer_XFPw · 2024-10-29

**Soundness:** 4
**Presentation:** 4
**Contribution:** 3
**Rating:** 6
**Confidence:** 5

**Summary:**

&nbsp;

The authors introduce a dynamic ensembling approach for Bayesian optimization considering both continuous and mixed continuous/discrete parameter spaces. A key finding of the authors' work appears to be that model selection (the limiting case of the authors' approach when the exponential moving average weighting parameter alpha is set to 1) outperforms weighted ensembling across the majority of experiments. The empirical findings of the paper are informative and rigorous and as such, I am leaning towards accepting the paper. I do, however, have some major concerns with the baselining of prior work in model selection for Bayesian optimization, namely the approach taken in [6] that I believe should be incorporated. Additionally, the work would be further strengthened if a thorough ablation on the components of HEBO could be performed. Most notably if the model ensembling inherent to HEBO could be decoupled from the ensembling approach the authors' introduce. I will upgrade my score if these concerns can be addressed during the rebuttal phase, albeit I understand it will require extensive work on the part of the authors.

&nbsp;

**Strengths:**

&nbsp;

1. The empirical evaluation of the authors' method is extensive and offers conclusive evidence that dynamic model selection outperforms a single surrogate model and a static ensemble across a suite of continuous and mixed continuous/discrete hyperparameter optimization tasks.

2. The authors codebase is well-documented and I am led to believe that the results are fully reproducible.

3. The paper is written in a clear and objective fashion.

4. The finding that model selection outperforms a weighted ensemble in the majority of experiments is a very interesting finding for the Bayesian optimization community.

&nbsp;

**Weaknesses:**

&nbsp;

**MAJOR POINTS**

&nbsp;

1. In Equation 6, the authors take the average of the individual model standard deviations to compute the ensemble standard deviation. In Equation 9 of [10], the authors provide an expression for the predictive uncertainty of the ensemble that decomposes uncertainty into aleatoric and epistemic uncertainty. As far as I can tell, Equation 6 is only computing the aleatoric uncertainty and is missing the term for epistemic uncertainty (disagreement between models). What was the authors' justification for omitting the epistemic uncertainty component of the predictive uncertainty?

2. It would be worth adding the work from [6] as an additional baseline method since the parameter setting of alpha=1 i.e. model selection appears to perform best in experiments. This method, which uses Bayesian optimization in an inner loop to select the best model at each iteration of BO would be applicable to experiments over continuous hyperparameter spaces. It would be informative to see how model selection using the MSE compares against model selection using the marginal likelihood as in [6].

3. For the empirical results in the appendix it may be worth plotting the log regret in place of the regret of each method. This may provide a clearer picture of the performance deltas to be expected between the methods.

4. In Section 6, the baseline single GP implementation is presumably implemented using HEBO? If so, it makes some sense that the single GP variant of HEBO is performant in the low sample regime as early in the optimization trace, the treatment of the GP hyperparameters is fully Bayesian and hence there is an inherent notion of model ensembling. It would be interesting to run an ablation with the robust acquisition formulation (fully Bayesian treatment of hyperparameters) turned off. This could be achieved by using only the input/output warping and MACE components of HEBO. This would also decouple the inherent model ensembling within HEBO from the ensembling introduced by the authors.

5. It would be worth emphasizing in the introduction as well as the overall narrative of the paper, the result that alpha=1 i.e. model selection outperforms ensembling in the experiments. This seems to be a key finding of the paper.

&nbsp;

**MINOR POINTS**

&nbsp;

1. There are some missing capitalizations in the references e.g. "Bayesian" in place of "bayesian".

2. When citing Bayesian optimization on line 47, it may be worth referencing some of the originating papers [2, 3] as discussed in [4].

3. On line 84, the statement that Gaussian processes are performant in settings where the dimension is lower than 20 should potentially be moderated in light of recent work such as [5] which shows that Bayesian optimization can be performant in problems with 100s of dimensions if the lengthscale prior is scaled with the dimensionality.

4. In light of the major points above, the sentence, "We present, for the first time, a dynamic ensembling approach for surrogate models in the context of HPO" may need to be moderated in light of [6] which introduces an ensemble of GP models (cf. Equation 8 in [6]) for Bayesian optimization and reports experimental results for hyperparameter tuning (neural networks and SVMs) of models fit on UCI datasets.

5. In the codebase it may be beneficial to rename the HEBO directory since I initially assumed this was directly copied from the HEBO codebase whereas in fact it contains code relevant for the method introduced in this paper.

6. On line 134, why is the loss function "estimated"?

7.  Line 139, see point 2 above.

8. On line 142, a set notation with curly braces may be more appropriate than parentheses.

9. On line. 154, is there a typo with the bolding of lambda in the correlation vector?

10. When citing the expected improvement acquisition function [9] should be cited as discussed in [4].

11. On line 217, in terms of the claim that this work introduces dynamic model ensembling for hyperparameter optimization see again the major points above, namely the reference to [6].

12. Line 230, extraneous colon.

13. Missing full stop at the end of Equation 16.

14. Figure 9, extraneous bracket around panel d. It may be worth additionally explaining the black dotted line in the figure as the distance to the closest variance bound.

15. In line 3 of Algorithm 1 it would be worth specifying how the model weights are initialized.

16. The notation in line 5 of Algorithm 1 is a little confusing. For example mu and sigma should be vector quantities and the notation suggests the models are fit on lambda alone and not the associated costs.

17. On line 7, it may be better if lambda_t is defined as a vector. Similarly on line 8.

18. On line 307, "heteroscedasticity and non-stationarity respectively" may be a better phrasing given that the Box-Cox and Yeo-Johnson transforms in HEBO address heteroscedasticity specifically whereas the Kumaraswamy transform addresses non-stationarity specifically. The authors indeed emphasize this in Appendix C.

19. In Section 4, in the description of the aspects of HEBO that yield performance gains, there are two additional components worthy of mention. The first, is the fact that HEBO uses the Multi-objective ACquisition Ensemble (MACE) method of [11] which ensembles the EI, PI, and UCB acquisition functions. The second, is that HEBO uses a robust acquisition function formulation which is equivalent to a fully Bayesian approach to the GP hyperparameters early in the optimization trace. In the HEBO paper, all 4 components were shown to improve performance independently of each other in an ablation study.

20. It would be worth citing the MACE paper [11] as the source work for the approach taken in HEBO. Out of interest, does the reference by Forrester also include such an acquisition function ensemble?

21. In the caption of Figure 2, it would be worth adding the number of random trials the errorbars are computed over.

22. Missing full stop at the end of Equation 9.

23. On line 784, I believe noise may still be heteroscedastic, yet adhere to a Gaussian noise model, save for the fact that the parameters of the Gaussian distribution will vary depending on the position in the input space? Willing to discuss this point further!

24. Equation 10 may be more clearly defined as a piecewise function?

25. It would be worth citing the Adam optimizer [12] given that it is used.

26. In line 845, the Q-function (state-action value function) holds the predicted discounted future return (discounted cumulative reward) rather than the expected simple reward.

27. For the significance test in Section I of the Appendix it would be good to formally state the null hypothesis and report the p-value of the permutation test.

&nbsp;

**REFERENCES**

&nbsp;

[1] Brown, T.B., Mann, B., Ryder, N., Subbiah, M., Kaplan, J., Dhariwal, P., Neelakantan, A., Shyam, P., Sastry, G., Askell, A. and Agarwal, S., 2020, December. [Language models are few-shot learners](https://proceedings.neurips.cc/paper/2020/file/1457c0d6bfcb4967418bfb8ac142f64a-Paper.pdf?ref=hackernoon.com). In Proceedings of the 34th International Conference on Neural Information Processing Systems (pp. 1877-1901).

[2] Kushner, HJ., [A Versatile Stochastic Model of a Function of Unknown and Time
Varying Form](https://www.sciencedirect.com/science/article/pii/0022247X62900112). Journal of Mathematical Analysis and Applications 5(1):150–167. 1962.

[3] Kushner HJ., [A New Method of Locating the Maximum Point of an Arbitrary Multipeak Curve in the Presence of Noise](https://asmedigitalcollection.asme.org/fluidsengineering/article-abstract/86/1/97/392213/A-New-Method-of-Locating-the-Maximum-Point-of-an?redirectedFrom=fulltext). Journal of Basic Engineering 86(1):97–106. 1964.

[4] Garnett, R., [Bayesian optimization](https://bayesoptbook.com/). Cambridge University Press. 2023.

[5] Hvarfner, C., Hellsten, E.O. and Nardi, L. [Vanilla Bayesian Optimization Performs Great in High Dimensions](https://proceedings.mlr.press/v235/hvarfner24a.html). Proceedings of the 41st International Conference on Machine Learning, in Proceedings of Machine Learning Research 235:20793-20817. 2024.

[6] Malkomes, G. and Garnett, R., [Automating Bayesian optimization with Bayesian optimization](https://proceedings.neurips.cc/paper_files/paper/2018/file/2b64c2f19d868305aa8bbc2d72902cc5-Paper.pdf). Advances in Neural Information Processing Systems, 31. 2018.

[7] Malkomes, G., Schaff, C. and Garnett, R., [Bayesian optimization for automated model selection](https://proceedings.neurips.cc/paper/2016/hash/3bbfdde8842a5c44a0323518eec97cbe-Abstract.html). Advances in Neural Information Processing Systems, 29. 2016.

[8] Gardner, J., Guo, C., Weinberger, K., Garnett, R. and Grosse, R.,[Discovering and exploiting additive structure for Bayesian optimization](https://proceedings.mlr.press/v54/gardner17a.html). In Artificial Intelligence and Statistics (pp. 1311-1319). PMLR. 2017.

[9] Saltines, VR., One Method of Multiextremum Optimization. Avtomatika i Vychislitel’naya Tekhnika (Automatic Control and Computer Sciences) 5(3):33–38. 1971.

[10] Kendall, A. and Gal, Y., [What uncertainties do we need in bayesian deep learning for computer vision?](https://proceedings.neurips.cc/paper/2017/hash/2650d6089a6d640c5e85b2b88265dc2b-Abstract.html). Advances in Neural Information Processing Systems, 30. 2017.

[11] Lyu, W., Yang, F., Yan, C., Zhou, D. and Zeng, X., [Batch Bayesian optimization via multi-objective acquisition ensemble for automated analog circuit design](https://proceedings.mlr.press/v80/lyu18a/lyu18a.pdf). In International Conference on Machine Learning (pp. 3306-3314). PMLR. 2018.

[12] Kingma and Ba, [Adam: A Method for Stochastic Optimization](https://arxiv.org/abs/1412.6980), ICLR 2015.

&nbsp;

**Questions:**

&nbsp;

1. In the introduction could the authors explain why the citation to "Language Models are few-shot learners" is an appropriate citation for the point that evaluating a single hyperparameter configuration of a model can be very expensive?

2. What is the motivation for assessing MSE only on the newly sampled points and not on the entire dataset collected so far in the trace? Do the authors have any intuition for how behavior would change if the MSE was defined on the full dataset? The motivation is outlined partially in Section C.2 of the Appendix.

&nbsp;

**Details Of Ethics Concerns:**

&nbsp;

No ethical concerns.

&nbsp;

---

> ### Author Response · Authors · 2024-11-24
> **Response to Reviewer XFPw**
>
> We thank the reviewer for their time and the helpful feedback. We appreciate the reviewer’s concerns and we address the comments below in order of appearance. We will upload the revised version shortly.
>
> # Responses to major weaknesses (1-5) and questions (6-7):
> * Predictive uncertainty: we agree that the total predictive variance should also contain the term for epistemic uncertainty, which is something we realised after submitting the paper but is missing in the present version. We will make necessary adjustments in the camera-ready version, although we do not expect the performance of DensBO to drastically change with the new uncertainty formulation.
> * Baseline based on “Automating BO with BO”: we thank the reviewer for drawing our attention to this missing reference. We will properly acknowledge this work and discuss the ways in which our method differs from it in the revised version of the paper; however, we fear we are time-constrained for including the mentioned approach as a baseline, one of the reasons being that the source code is in MATLAB, which in this particular situation requires more overhead for setting up additional experiments, and as such we leave this comparison for future work.
> * Log-regret plots: we will add log-regret plots for the empirical results in the appendix of the revised version.
> * Single-GP HEBO ablation without robust formulations of acquisition functions: we agree with the reviewer that turning off Bayesian treatment of GP hyperparameters is a point worthy of further investigation, and considering a single-GP baseline equipped with only input/output warping and MACE would allow to better capture performance gains achieved by our dynamic model ensembling. However, due to limited time we will leave this ablation study for the camera-ready version of the paper.
> * Model selection vs. ensembling: while it is true that model selection (\alpha=1) exhibits strong performance across most experimental settings, for expensive target functions (more than 1h of mean time for evaluating the function) it seems to be beneficial to retain some small degree of model accuracy history (\alpha=0.9). We will make sure to emphasise the key takeaways accordingly in the revised version.
> * Reference in the introduction: thank you for pointing this out. This reference might be misleading, and will be replaced by references to https://arxiv.org/pdf/2203.15556 and https://arxiv.org/pdf/2306.08055 (and references within). These papers only serve to reinforce the (extreme) case where performing HPO of a model is crucial for good performance, and training and evaluating a model in question (here: deep neural networks) is prohibitively expensive.
> * Motivation for assessing MSE on newly sampled points (“local” accuracy):  evaluating the accuracy of the competing models on the points generated through the optimisation of the acquisition function in the current iteration, rather than on an externally generated test set, allows us to assess the “local” accuracy of the models. This approach aligns particularly well with the concept of surrogate-based optimisation rather than traditional machine learning prediction. In surrogate-based optimisation, we do not require a model that is globally accurate across the entire search space. Instead, we need a surrogate that performs well in specific areas of interest, such as basins of attraction. We argue that focusing on local accuracy is a key component for efficient model selection in our ensembling method. However, as the assignment of weights to the competing models is based on their accuracy on newly sampled points, this could tend to favour models that predict lower variance. This may lead to a more exploitative tendency in the BO algorithm. Therefore, we plan to conduct further experiments to compare our method against baselines that directly address the exploration-exploitation trade-off in BO. Regarding the behaviour of the dynamic ensembling based on MSE on full dataset, we did conduct preliminary experiments precisely in this setting, which we did not report in the paper as the performance of ensembling using the weighting scheme defined on MSE on full dataset observed so far was not satisfactory; in fact, dynamic ensembling rarely outperformed the single-surrogate baselines in this case, all else being equal.

---

> > ### Author Response · Authors · 2024-11-24
> > **Response to Reviewer XFPw (cont'd)**
> >
> > # Responses to minor points (numbering in order of appearance in the review):
> > If not explicitly answered below, we thank the reviewer for a sharp eye and attention to detail that allowed for not only catching typos (points 12, 13, 14, 22), but also for pointing out parts that require additional clarification. In light of this, we will make necessary adjustments concerning references (style and missing citations, points 1, 2, 7, 10, 20, 25); we will polish mathematical notation, more precisely define concepts we talk about throughout the paper, as well as provide additional explanations and information where it is missing, e.g., the number of trials for plotting the error bars,  a formal statement of the null hypothesis and p-values for statistical significance, or more precise description of HEBO components and renaming the HEBO codebase in our repository, to name a few (points 5, 6, 8, 9, 14, 15, 16, 17, 18, 19, 21, 24, 26, 27).
> >
> > * Point 3: We agree that the statement re: BO not performing well in high dimensions should be moderated. In fact, we will properly acknowledge the referenced paper and make adjustments in the revised version. However, we do not make any assumptions on the lengthscale prior; we rather argue that the dynamic ensembling is applicable “out-of-the-box”, regardless of the BO framework at hand (i.e., available surrogates and acquisition functions). We chose HEBO since it showed SOTA performance on HPO use-cases (see https://datasets-benchmarks-proceedings.neurips.cc/paper/2021/file/93db85ed909c13838ff95ccfa94cebd9-Paper-round2.pdf), however in preliminary experiments we tested the same DensBO approach within SMAC, which we do not report in the paper for reasons of space and coherent narrative, but which allow to derive very similar conclusions: all variants of dynamic ensembling with different \alpha values outperformed single-surrogate BO in SMAC.
> > * Point 4: Again, we agree that the statement should be moderated in light of the work in the mentioned reference. However, in this work the authors consider only ensembles of different GPs, whereas we consider a mixed ensemble of GP and tree-based techniques.
> >
> > We hope that the responses clarify all raised concerns and we remain available for any additional questions or comments. If there is anything in particular that would allow for improving the score, we would very much appreciate it if you could let us know.

---

> > ### Comment · Reviewer_XFPw · 2024-11-25
> > **Many Thanks to the Authors for their Response**
> >
> > &nbsp;
> >
> > **[P1] Predictive uncertainty**
> >
> > &nbsp;
> >
> > As I understand the expression for the predictive uncertainty also affects the experiments?
> >
> > &nbsp;
> >
> > **[P2] Automating BO with BO baseline**
> >
> > &nbsp;
> >
> > Many thanks for pointing out that the code is in MATLAB. This indeed makes a direct comparison more challenging.
> >
> > &nbsp;
> >
> > **[P3] Assessing MSE on Newly Sampled Points**
> >
> > &nbsp;
> >
> > I fully agree with the authors' point about the respective goals of active learning/experimental design (global accuracy) vs. Bayesian optimization (local accuracy around the optimum). To this end it may be worth experimenting with weighted MSE with a bias towards points with favorable label values y. Albeit it is a priori unclear how the weighting should be scheduled as a function of the number of iterations in the optimization trace in terms of its impact on the exploration/exploitation tradeoff.
> >
> > &nbsp;
> >
> > **__SUMMARY__**
> >
> > &nbsp;
> >
> > Synthesizing the concerns of the other reviewers, the main points of concern at the moment seem to be the implementation of the experiments namely:
> >
> > &nbsp;
> >
> > 1. The correction to the predictive uncertainty (Me).
> > 2. Comparison against static ensembling methods and other surrogate ensemble methods (Reviewer mBTA).
> > 3. Comparison against methods that ensemble different BO approaches (Reviewer dmp5).
> >
> > &nbsp;
> >
> > If the empirical study can be expanded to address these points I would be inclined to increase my score. I understand, however, that running these experiments may be infeasible in the time remaining for the rebuttal phase. I believe that this work has promise and I would encourage the authors to attempt to get feedback from the current reviewers on any experiments they may be able to run. This feedback will hopefully be useful for the next conference review cycle.
> >
> > &nbsp;

---

> > > ### Author Response · Authors · 2024-12-01
> > > **Planned revisions**
> > >
> > > We sincerely thank the reviewers for their constructive feedback and for their overall positive reception of the potential of our approach. As pointed out by Reviewer XFPw, it was unfortunately not feasible for us to conduct all the requested experiments within the limited time of the rebuttal phase. However, we commit to including the following in the camera-ready version: (1) additional details on epistemic uncertainty, (2) a comparison with static and other ensembling methods, and (3) a comparison with methods that ensemble different BO approaches. We completely understand if the reviewers decide not to adjust their scores based on this commitment on our side, and we are deeply grateful for their feedback regardless of the outcome. We strongly believe the suggested improvements will enhance the quality of our work and facilitate a smoother review process in future submissions.
> > >
> > > We further elaborate on several points raised in subsequent discussion below.
> > >
> > > * Epistemic uncertainty:
> > > In light of the discussion about epistemic uncertainty, we need to further clarify this point by taking a step back into our preliminary experiments, as there were some settings for which we did not get satisfactory performance of the dynamic ensembling, leading us to move away from them and thus those experiments ended up not being reported in the paper.
> > > One such setting concerns a way of computing the variance as done in (Hutter et al 2014, https://www.sciencedirect.com/science/article/pii/S0004370213001082), section 4.3.2. This was among the first things we attempted, which did not work well, and that idea was soon discarded. Upon careful reflection, this equation corresponds exactly to the equation 9 in https://proceedings.neurips.cc/paper_files/paper/2017/file/2650d6089a6d640c5e85b2b88265dc2b-Paper.pdf (that Reviewer XFPw pointed to). This of course means that we need to very carefully discuss this in the paper and we acknowledge the confusion that arose regarding it. However, to this end we do not yet have any explanations as to why it did not yield good performance, and this is something worthy of investigating in more detail. We will repeat those experiments for the camera-ready version of the paper, as we regrettably did not store those results.
> > >
> > > * Leave-one-instance-out cross-validation as a baseline:
> > > Thank you for pointing this out. We will add this as a baseline along with other baselines for the camera-ready version of the paper (i.e., virtual best surrogate per problem, “GP if only continuous, RF otherwise”, and “HEBO with only input/output warping and MACE turned on”).
> > >
> > > * Sequential setting of HEBO:
> > > The parallel batched setting of HEBO was used consistently for all experiments, regardless of the underlying surrogate model, because it allowed us to reduce the overall waiting time for the results, as we were lucky to have access to enough compute that would enable parallelisation. We argue that our observations will still hold in the case of sequential execution, and we will add the relevant experiments in the camera-ready version of the paper.
> > >
> > > * Comparison to other methods (static ensembling, different surrogates, different BO approaches):
> > > We were able to identify, with the help of the reviewers that raised those points, various methods to extend our comparison, which will render the paper more complete. We will include the experiments concerning approaches from relevant literature (where technically possible) oin the camera-ready version of the paper.

---

> > > > ### Comment · Reviewer_XFPw · 2024-12-02
> > > > **Many Thanks to the Reviewers for their Response**
> > > >
> > > > &nbsp;
> > > >
> > > > Many thanks to the reviewers for their response and for pointing me towards the Hutter et al. 2014 reference. The uncertainty decomposition is indeed quite interesting. One thought that occurred to me is that the epistemic uncertainty derived from the disagreement between constituent models of the ensemble assumes that the predictions of those models are independent? In Kendall et al. 2017, the constituent "models" represent samples from a posterior distribution. In contrast, the authors' ensembling approach appears to involve more general constituent models such as GPs with different kernels. Correlated predictions may undermine the fidelity of the epistemic uncertainty estimates but I don't have a good intuition for how this would play out empirically.
> > > >
> > > > &nbsp;

---

### Official Review · Reviewer_mBTA · 2024-11-03

**Soundness:** 2
**Presentation:** 3
**Contribution:** 2
**Rating:** 3
**Confidence:** 4

**Summary:**

The authors present DensBO, a method which dynamically uses a weighted ensemble of different surrogate models when optimizing the acquisition function.

**Strengths:**

- The paper is well-written and the method is clearly presented.
- The ablation with the smoothing factor $\alpha$ is interesting and provides insight into how much the prior performance should impact the ensemble. Furthermore, the promising performance of $\alpha=1.0$ alleviates some of the overhead cost of using an ensemble surrogate model.
- The authors conduct an extensive empirical evaluation on over 800 optimization problems and demonstrate that their ensemble method consistently outperforms the single models as well as the static ensemble. Furthermore, the ensemble is more effective than individual models when controlling for the total budget as well as controlling for the number of evaluations.
- The experiments are described in detail and the results seem fully replicable.

**Weaknesses:**

- This paper does not provide enough evidence that ensembles, and the additional costs associated with fitting multiple models, outperform careful initial model selection across the many different objectives. It would be interesting to see another baseline for “performance of best single-surrogate model for problem”, which would allow us to compare the impact of ideal model selection compared to ensembling. For a more realistic setting, we could also include another baseline of “performance of best single-surrogate model as determined by best MSE on initial points.” If this baseline performs well, this indicates that the overhead cost of ensembles are unnecessary.
- Furthermore, there is no comparison between DensBO vs other static ensembling methods which carefully select the ensemble model weights (rather than using a naive equal weighting). It is unclear how much of the improved performance originates from the dynamic ensembling proposed from the paper or from ensembling in general.
- While I believe DensBO is the first to use a dynamic ensemble of GP and trees fit to the same BO optimization trajectory, there has been previous work which utilizes dynamic ensembling for BO which are weighted based on model fit [1, 2, 3]. If the claim is that the specific MSE weighting scheme is desirable, it would be helpful to see comparisons with other weighting schemes.

1. Feurer et al, Scalable Meta-Learning for Bayesian Optimization using Ranking-Weighted Gaussian Process Ensembles, 2018.
2. Lu et al, Ensemble Gaussian Processes with Spectral Features for Online Interactive Learning with Scalability, 2020.
3. Nagy et al, Ensemble Gaussian Processes for Adaptive Autonomous Driving on Multi-friction Surfaces, 2023

**Questions:**

- Have you tried running experiments with a different number of starting points? Due to the varying scaling properties of the individual surrogate models, the number of initial points may impact the performance of the methods given a fixed budget. Furthermore, considering your hypothesis about why GB works poorly in line 462, it may be helpful to understand the performance gap.

---

> ### Author Response · Authors · 2024-11-24
> **Response to Reviewer mBTA**
>
> We thank the reviewer for their time and the helpful feedback. We appreciate the reviewer’s concerns and we address the comments below in order of appearance. We will upload the revised version shortly.
>
> * Baselines and model selection: we will include the comparison with the “best surrogate per problem” (as virtual best surrogate) in the revised version. The baseline “best surrogate as determined by MSE on initial points” (to justify the overhead cost of ensembles) will not hold in case of GP, as GP is first fit on the initial points and will recover them without loss due to interpolation.
> * Static ensembles and weight selection: existing work on static ensembles considers either different surrogate model portfolios or different methodology of building the ensemble, rendering a fair comparison impossible. It is known that ensembles allow for performance gains across different machine learning tasks, which is a fact that motivates our methodology. We argue that the performance of our approach is further enhanced by dynamically focusing on the “local” accuracy of the surrogate models, i.e., the accuracy on the newly sampled points previously unseen by the model, since from a classical optimisation viewpoint we are interested in those surrogates which perform well in specific areas of interest (such as basins of attraction) rather than overall good surrogates.
> * Related work on dynamic ensembling: we will properly acknowledge and discuss the provided references in the revised version.
> * Varying the DoE size: we exploit the fact that BO is designed for low-budget, expensive-to-evaluate settings in which a small size of DoE is preferable since the total budget is restrained. We will add an ablation study varying the DoE size for different small values of the initial design (16, 32) and measuring the impact on performance in the revised version of the paper.
>
> We hope that the responses clarify all raised concerns and we remain available for any additional questions or comments. If there is anything in particular that would allow for improving the score, we would very much appreciate it if you could let us know.

---

> > ### Comment · Reviewer_XFPw · 2024-11-25
> > **Best Surrogate as Determined by MSE on Initial Points**
> >
> > &nbsp;
> >
> > It may be possible to choose the best surrogate by applying leave-one-out cross-validation on the initial points. This would mitigate against the small sample size.
> >
> > &nbsp;

---

> > > ### Author Response · Authors · 2024-12-01
> > > **Planned revisions**
> > >
> > > We sincerely thank the reviewers for their constructive feedback and for their overall positive reception of the potential of our approach. As pointed out by Reviewer XFPw, it was unfortunately not feasible for us to conduct all the requested experiments within the limited time of the rebuttal phase. However, we commit to including the following in the camera-ready version: (1) additional details on epistemic uncertainty, (2) a comparison with static and other ensembling methods, and (3) a comparison with methods that ensemble different BO approaches. We completely understand if the reviewers decide not to adjust their scores based on this commitment on our side, and we are deeply grateful for their feedback regardless of the outcome. We strongly believe the suggested improvements will enhance the quality of our work and facilitate a smoother review process in future submissions.
> > >
> > > We further elaborate on several points raised in subsequent discussion below.
> > >
> > > * Epistemic uncertainty:
> > > In light of the discussion about epistemic uncertainty, we need to further clarify this point by taking a step back into our preliminary experiments, as there were some settings for which we did not get satisfactory performance of the dynamic ensembling, leading us to move away from them and thus those experiments ended up not being reported in the paper.
> > > One such setting concerns a way of computing the variance as done in (Hutter et al 2014, https://www.sciencedirect.com/science/article/pii/S0004370213001082), section 4.3.2. This was among the first things we attempted, which did not work well, and that idea was soon discarded. Upon careful reflection, this equation corresponds exactly to the equation 9 in https://proceedings.neurips.cc/paper_files/paper/2017/file/2650d6089a6d640c5e85b2b88265dc2b-Paper.pdf (that Reviewer XFPw pointed to). This of course means that we need to very carefully discuss this in the paper and we acknowledge the confusion that arose regarding it. However, to this end we do not yet have any explanations as to why it did not yield good performance, and this is something worthy of investigating in more detail. We will repeat those experiments for the camera-ready version of the paper, as we regrettably did not store those results.
> > >
> > > * Leave-one-instance-out cross-validation as a baseline:
> > > Thank you for pointing this out. We will add this as a baseline along with other baselines for the camera-ready version of the paper (i.e., virtual best surrogate per problem, “GP if only continuous, RF otherwise”, and “HEBO with only input/output warping and MACE turned on”).
> > >
> > > * Sequential setting of HEBO:
> > > The parallel batched setting of HEBO was used consistently for all experiments, regardless of the underlying surrogate model, because it allowed us to reduce the overall waiting time for the results, as we were lucky to have access to enough compute that would enable parallelisation. We argue that our observations will still hold in the case of sequential execution, and we will add the relevant experiments in the camera-ready version of the paper.
> > >
> > > * Comparison to other methods (static ensembling, different surrogates, different BO approaches):
> > > We were able to identify, with the help of the reviewers that raised those points, various methods to extend our comparison, which will render the paper more complete. We will include the experiments concerning approaches from relevant literature (where technically possible) oin the camera-ready version of the paper.

---

> > > > ### Comment · Reviewer_mBTA · 2024-12-02
> > > >
> > > > Thank you for your response! I think the method is interesting, and I'm looking forward to seeing future iterations of this work with more comprehensive empirical results.

---

### Official Review · Reviewer_2TAw · 2024-11-03

**Soundness:** 1
**Presentation:** 2
**Contribution:** 1
**Rating:** 3
**Confidence:** 5

**Summary:**

The authors propose a new surrogate model for Bayesian optimization. They use a weighted ensemble of 4 different surrogates which are weighted best on their prediction performance in the past. This new surrogate is compared to each individual surrogate on several benchmarks with respect to rank.

**Strengths:**

The paper is well-written and the idea clearly communicated. The paper uses sufficiently many benchmarks. The idea to dynamically choose the most suitable surrogate model during optimization is interesting.

**Weaknesses:**

The evaluation metric is not a suitable choice for this setup for two main reasons:

1. We have clusters of surrogates: there are 5 variants of ENS, 3 variants of tree-based methods and 1 GP method. Why is this a problem? Let's assume we want to compare GP to ENS on 2 benchmarks. If ENS is better than the GP on one benchmark and the GP better on the other, both would get a rank of 1.5. If we had 5 variants of ENS (where these variants also have a clear ranking among each other), then ENS-best would get a rank of 1 on the problem where ENS does better than GP and a rank of 2 where the GP does better. However, the GP gets a rank of 6 where the ENS variants do better. Thus, the GP overall gets a rank of 3.5 and ENS-best 1.5. Only adding more variants of ENS make the GP look much worse than it is. In my opinion, we should compare 1 ENS variant with a GP and RF (no other tree-based methods).
2. We don't see absolute differences: we see the GP doing initially very well for "cheap" benchmarks. The authors claim to eventually do better. However, differences towards the end of the optimization process might be completely meaningless (small absolute gains).

The approach to set the ensemble weights is not well motivated and not ablated (see "Questions" for ideas what to ablate). I do not understand why this depends on the answer to "which model would have been best in evaluating the current point?" if we continue choosing points somewhere else in the space.

Adding the state-of-the-art for the respective benchmarks would be useful to give a useful baseline for the benchmarks.

Missing baseline: use GP if the problem instance contains only continuous hyperparameters, otherwise use RF

**After review:** I wish the authors had added the missing baseline mentioned above. The authors said that this is computationally infeasible during the short time while they in fact have results for GP and RF on all problems. Therefore, it is not clear to me why a posthoc analysis isn't possible and this appears to be a red flag to me.

**Questions:**

How important is the initialization of w? What happens if we init all with equal weights?

How well does ENS with oracle w? This means that we choose only the surrogate among the available candidates that did best on the respective benchmark.

What is the motivation for using 3 tree-based models? How would ENS change if there was only RF (best tree-based model) and GP?

How important is it to choose weights based on fit to last observe point rather than general fit?

---

> ### Author Response · Authors · 2024-11-24
> **Response to Reviewer 2TAw**
>
> We thank the reviewer for their time and the helpful feedback. We appreciate the reviewer’s concerns and we address the comments below in order of appearance. We will upload the revised version shortly.
>
> * Evaluation metric – clustering of surrogates: we will include the comparison of one ENS variant with only RF and GP in the appendix of the revised version.
> * Evaluation metric – absolute differences: we agree with the reviewer that the mean rank plot is relative, however this is why we provide a table of critical differences (see Appendix I, Table 6) to support the claim that ENS indeed outperforms other methods in a statistically significant way. We will include the plots with normalised loss in the appendix of the revised version.
> * Setting the ensemble weights (“local” vs “global” surrogate model accuracy): evaluating the accuracy of the competing models on the points generated through the optimisation of the acquisition function in the current iteration, rather than on an externally generated test set, allows us to assess the “local” accuracy of the models. This approach aligns particularly well with the concept of surrogate-based optimisation rather than traditional machine learning prediction. In surrogate-based optimisation, we do not require a model that is globally accurate across the entire search space. Instead, we need a surrogate that performs well in specific areas of interest, such as basins of attraction. We argue that focusing on local accuracy is a key component for efficient model selection in our ensembling method. However, as the assignment of weights to the competing models is based on their accuracy on newly sampled points, this could tend to favour models that predict lower variance. This may lead to a more exploitative tendency in the BO algorithm. Therefore, we plan to conduct further experiments to compare our method against baselines that directly address the exploration-exploitation trade-off in BO.
> * Missing baselines: we will add a comparison of DensBO to the best surrogate per benchmark (virtual best surrogate) in the appendix in the revised version. The baseline “GP if continuous only, RF otherwise” would however require computational efforts that would make it hard to include it in time for the revised version; we can discuss this in the paper.
> * Weight initialisation: we will include an ablation study with all equal initial weights in the revised version of the appendix.
> * Oracle weights: a comparison of DensBO with a virtual best surrogate (see point 4) would allow to derive insights of how the dynamic ensembling approach would perform if the weight assignment was done in an oracle fashion.
> * Motivation behind 3 tree-based models: as GB and ET are also powerful on regression tasks, it was interesting to investigate how they would behave (separately and combined within an ensemble) as surrogates for BO on an HPO task. We will add an ablation study in the revised version of the paper to check this setting with only GP and RF (see point 1).
> * Choosing weights based on “local” accuracy: please see point 3.
>
> We hope that the responses clarify all raised concerns and we remain available for any additional questions or comments. If there is anything in particular that would allow for improving the score, we would very much appreciate it if you could let us know.

---

> ### Comment · Reviewer_2TAw · 2024-12-01
>
> > The baseline “GP if continuous only, RF otherwise” would however require computational efforts that would make it hard to include it in time for the revised version
>
> You ran GP and RF individually. It is not required to run any additional experiments. Simply aggregate the results in a different way.
>
> Same also applies for the comparison against oracle weights.
>
> > Evaluation metric
>
> I was able to find Table 6, but can you point me explicitly to where you present ENS with GP/RF only? In the current light, the statistical test is still not meaningful to me since it still suffers from the problems I've outlined in the review.
>
> >  In surrogate-based optimisation, we do not require a model that is globally accurate across the entire search space. Instead, we need a surrogate that performs well in specific areas of interest, such as basins of attraction.
>
> How do you define accurate? The mean prediction of a surrogate can only be accurate in specific areas of interest. However, if the uncertainty prediction is accurate, the model will not explore these areas at all. I'd argue that we actually require an "accurate" surrogate in the sense that it can confidently rule out regions. This is not possible with a model that is performing poor in some regions. Inaccurate can mean over- but also underestimating performance.

---

### Official Review · Reviewer_dmp5 · 2024-11-04

**Soundness:** 2
**Presentation:** 3
**Contribution:** 3
**Rating:** 5
**Confidence:** 4

**Summary:**

DensBO offers a novel and dynamic approach to ensemble surrogate models in Bayesian optimization. While some transfer knowledge between tasks have been used in the past, this method uses the dynamic ensemble on the same task. This allows the use of different surrogate models at different stages of the optimization. This might also be useful to deal with numerical and categorical search spaces.
The method trains all surrogate models and creates a weighted ensemble, where the weights are updated using an exponential moving average between the previous and the new weights. At each time step, the surrogate model that has the lowest MSE error on the new sampled points, gets the weight of 1 and zero for the rest.
The paper evaluates the method on YAHPO Gym and JAHS-Bench-201 making use of a total of 859 available instances. The results show that the proposed method outperforms using individual models in Bayesian optimization. While the method comes at the cost of an additional hyperparameter, the paper convincingly demonstrates that the method is very robust as long as the hyperparameter is set to a high value.
Overall, the paper proposes an interesting addition to the Bayesian optimization methodology, that is of general interest to the ICLR, but has one major weakness (see below) due to which I have to rate this paper borderline reject.

**Strengths:**

The paper is easy to read and understand.
* The paper explains the choices of the surrogate model by DensBO at the different stages of the optimization. Although gradient boosting (GB) is not competitive as a single surrogate model, the authors show that their model can pick random forest (RF)in the beginning of the optimization (few observations) and then dynamically chooses GB when they have a higher number of evaluations.
*  They also show results on different budgets, which is very interesting to point out when the DensBO is not working well. In a small budget Gaussian processes outperform DensBO. In fact, ensembling requires training and querying more than one surrogate model, which means that it increases the overhead of BO, and should thus not be used in very low budget settings.
* Finally, the explanation of the fraction budget used as a hyper parameter. The authors test and explain its effect on the behavior of DensBO.

**Weaknesses:**

* The experimental setup is uncommon for Bayesian optimization and might introduce unwanted bias into the evaluation. Concretely, the models are only updated every eight iterations (see line 819), and references the HEBO documentation and SMAC. However, I cannot find a clear recommendation to do so in the HEBO documentation, and also no trace of this in the SMAC paper [1]
* Related work does not discuss meta-learning in Bayesian optimization, see for example the work by Martin Wistuba and Nicolas Schilling between 2015 and 2018. These are probably not helpful for solving the problem at hand, but at least should be mentioned.
* Additional solutions to the 2021 NeurIPS BBO challenge besides HEBO that ensemble different BO methods are not discussed. In particular, the 2nd place, and the winner of the warm start leaderboard, used extensive ensembling of different BO methods (2nd place) and surrogates (winner of the warm start leaderboard, see also [2]).

References:
1. https://jmlr.csail.mit.edu/beta/papers/v23/21-0888.html
2. https://arxiv.org/abs/2012.08180

**Questions:**

* The fact that GPs do not work well on dimensions above 20 has been disputed recently [1, 2]. Also, the provided references about BO with GPs not working for a higher number of dimensions are 7 years old or older. Can you still make this claim using up-to-date-literature?
* Scikit-learn’s implementation of random forests and extremely randomized trees does not handle categorical features - how do you deal with this in practice? (The same might hold for gradient boosting, depending on the scikit-learn class used)
* How is the uncertainty for gradient boosting computed?

References:
1. https://arxiv.org/abs/2402.02229
2. https://arxiv.org/abs/2402.02746

---

> ### Author Response · Authors · 2024-11-24
> **Response to Reviewer dmp5**
>
> We thank the reviewer for their time and the helpful feedback. We appreciate the reviewer’s concerns and we address the comments below in order of appearance. We will upload the revised version shortly.
>
> * Experimental setup: in fact, retraining after 8 iterations seems to be the best practice in the relevant frameworks. The hyperparameter ‘retrain_after’ is set to 8 by default in SMAC3 (see the API https://automl.github.io/SMAC3/main/api/smac.main.config_selector.html#smac.main.config_selector.ConfigSelector; although it was never explicitly recommended to do so in the paper itself, a prevalent number of HPO studies that use SMAC3 do not deviate from using retrain_after set to 8). For HEBO, this is also true (see the HEBO tutorial example here: https://hebo.readthedocs.io/en/latest/optimisation.html). Section near the end, batched HEBO uses 8 points per iteration. We hope this clarifies the decision of batch size 8.
> * Related work: indeed, we do not properly acknowledge or discuss the cited papers, both on using GP transfer surrogate-based ensembles (Wistuba and Schilling) and on the runner-ups of the NeurIPS BBO competition. We will discuss them in related work in the revised version.
> * Claim that GPs do not work well in high dimensions: in light of recent results that suggest otherwise, we do not make any assumptions on the lengthscale, while the reason why vanilla BO performs well is scaling the lengthscale prior with dimensionality. We rather argue that, whatever BO-based framework we use, the dynamic ensembling methodology will still transfer well, regardless of which surrogate models we have available in the framework and their inner mechanisms. However, we will make this clearer in the revised version, acknowledging recent results in this direction.
> * Categorical features: we use ordinal encoding to deal with categorical features, as done in HEBO (see https://github.com/huawei-noah/HEBO/blob/master/HEBO/hebo/design_space/categorical_param.py).
> * Gradient boosting uncertainty: the uncertainty for gradient boosting is computed based on Equation 3 in Section 2.2 (variance for tree-based models), where the total variance is defined based on the variance of the predictions of the leaves (inspired by Hutter et al. 2011: https://ml.informatik.uni-freiburg.de/wp-content/uploads/papers/11-LION5-SMAC.pdf where the authors calculate the predictive mean and variance of random forest in Section 4.1).
>
> We hope that the responses clarify all raised concerns and we remain available for any additional questions or comments. If there is anything in particular that would allow for improving the score, we would very much appreciate it if you could let us know.

---

> > ### Comment · Reviewer_dmp5 · 2024-11-25
> > **Thank you very much for your detailed answer.**
> >
> > > Experimental setup: in fact, retraining after 8 iterations seems to be the best practice in the relevant frameworks. The hyperparameter ‘retrain_after’ is set to 8 by default in SMAC3 (see the API https://automl.github.io/SMAC3/main/api/smac.main.config_selector.html#smac.main.config_selector.ConfigSelector; although it was never explicitly recommended to do so in the paper itself, a prevalent number of HPO studies that use SMAC3 do not deviate from using retrain_after set to 8). For HEBO, this is also true (see the HEBO tutorial example here: https://hebo.readthedocs.io/en/latest/optimisation.html). Section near the end, batched HEBO uses 8 points per iteration. We hope this clarifies the decision of batch size 8.
> >
> > Thank you very much for pointing me to the relevant piece of information in the SMAC docs. Looking into the variable `retrain_after` in the docs, I found [this page](https://automl.github.io/SMAC3/v2.2.0/advanced_usage/12_optimizations.html), which suggests to to change this variable to 1 for high target function evaluation times. Now the question is: "what is high"? I would assume that for machine learning we can always assume that we are in the "high" regime, because machine learning models are often costly to evaluate.
> >
> > Following up on this, I also looked at the code submitted with the paper, and found that it does not refit the HEBO model after eight iterations as mentioned in the paper, but that it runs HEBO in a synchronous parallel fashion, similar to the NeurIPS BBO challenge. This makes the results highly inconclusive in my opinion, and I will not raise my score unless the authors add results for the standard, sequential setting. In addition, I suggest clarifying the experimental setup and mentioning in the main paper that the results are only valid for the parallel setting.
> >
> > > Gradient boosting uncertainty: the uncertainty for gradient boosting is computed based on Equation 3 in Section 2.2 (variance for tree-based models), where the total variance is defined based on the variance of the predictions of the leaves (inspired by Hutter et al. 2011: https://ml.informatik.uni-freiburg.de/wp-content/uploads/papers/11-LION5-SMAC.pdf where the authors calculate the predictive mean and variance of random forest in Section 4.1).
> >
> > This needs to be added to the paper. I was expecting that you use quantile regression and a quantile regression acquisition function here, as for example, done in [scikit-optimize](https://scikit-optimize.github.io/stable/modules/generated/skopt.gbrt_minimize.html#skopt.gbrt_minimize), a [recent ICML paper](https://proceedings.mlr.press/v202/salinas23a.html), or a [somewhat older workshop paper](https://arxiv.org/abs/2101.02289).
> >
> > > However, we will make this clearer in the revised version
> >
> > Thank you very much for being open wrt improving the related work section.
> >
> > Here are some answers related to other reviews:
> >
> > > It may be possible to choose the best surrogate by applying leave-one-out cross-validation on the initial points. This would mitigate against the small sample size.
> >
> > I think this would be an important baseline / comparison.
> >
> > > Predictive uncertainty: we agree that the total predictive variance should also contain the term for epistemic uncertainty, which is something we realised after submitting the paper but is missing in the present version. We will make necessary adjustments in the camera-ready version, although we do not expect the performance of DensBO to drastically change with the new uncertainty formulation.
> >
> > A comparison of these two formulations would be very interesting. I do not agree with reviewr XFPw that this is a "correction" as we do not know which one is better, and incorporating the disagreement between the models might just increase overall uncertainty and decrease the final optimization performance.

---

> > > ### Comment · Reviewer_XFPw · 2024-11-25
> > > **Predictive Uncertainty**
> > >
> > > &nbsp;
> > >
> > > In response to Reviewer dmp5, "correction" is perhaps too strong a term. The decomposition of the predictive uncertainty into aleatoric and epistemic uncertainty (model disagreement) would appear to be the most principled approach but I agree that it is unclear how this would play out empirically.
> > >
> > > &nbsp;

---

> > > ### Author Response · Authors · 2024-12-01
> > > **Planned revisions**
> > >
> > > We sincerely thank the reviewers for their constructive feedback and for their overall positive reception of the potential of our approach. As pointed out by Reviewer XFPw, it was unfortunately not feasible for us to conduct all the requested experiments within the limited time of the rebuttal phase. However, we commit to including the following in the camera-ready version: (1) additional details on epistemic uncertainty, (2) a comparison with static and other ensembling methods, and (3) a comparison with methods that ensemble different BO approaches. We completely understand if the reviewers decide not to adjust their scores based on this commitment on our side, and we are deeply grateful for their feedback regardless of the outcome. We strongly believe the suggested improvements will enhance the quality of our work and facilitate a smoother review process in future submissions.
> > >
> > > We further elaborate on several points raised in subsequent discussion below.
> > >
> > > * Epistemic uncertainty:
> > > In light of the discussion about epistemic uncertainty, we need to further clarify this point by taking a step back into our preliminary experiments, as there were some settings for which we did not get satisfactory performance of the dynamic ensembling, leading us to move away from them and thus those experiments ended up not being reported in the paper.
> > > One such setting concerns a way of computing the variance as done in (Hutter et al 2014, https://www.sciencedirect.com/science/article/pii/S0004370213001082), section 4.3.2. This was among the first things we attempted, which did not work well, and that idea was soon discarded. Upon careful reflection, this equation corresponds exactly to the equation 9 in https://proceedings.neurips.cc/paper_files/paper/2017/file/2650d6089a6d640c5e85b2b88265dc2b-Paper.pdf (that Reviewer XFPw pointed to). This of course means that we need to very carefully discuss this in the paper and we acknowledge the confusion that arose regarding it. However, to this end we do not yet have any explanations as to why it did not yield good performance, and this is something worthy of investigating in more detail. We will repeat those experiments for the camera-ready version of the paper, as we regrettably did not store those results.
> > >
> > > * Leave-one-instance-out cross-validation as a baseline:
> > > Thank you for pointing this out. We will add this as a baseline along with other baselines for the camera-ready version of the paper (i.e., virtual best surrogate per problem, “GP if only continuous, RF otherwise”, and “HEBO with only input/output warping and MACE turned on”).
> > >
> > > * Sequential setting of HEBO:
> > > The parallel batched setting of HEBO was used consistently for all experiments, regardless of the underlying surrogate model, because it allowed us to reduce the overall waiting time for the results, as we were lucky to have access to enough compute that would enable parallelisation. We argue that our observations will still hold in the case of sequential execution, and we will add the relevant experiments in the camera-ready version of the paper.
> > >
> > > * Comparison to other methods (static ensembling, different surrogates, different BO approaches):
> > > We were able to identify, with the help of the reviewers that raised those points, various methods to extend our comparison, which will render the paper more complete. We will include the experiments concerning approaches from relevant literature (where technically possible) oin the camera-ready version of the paper.

---

> > > > ### Comment · Reviewer_dmp5 · 2024-12-02
> > > > **Looking forward to the updated version**
> > > >
> > > > Thank you very much for your answers and openness for additional experiments. I believe the the promised updates and outcomes of the discussion will make this an excellent paper. Without the updates (or a sign of updates in an updated PDF) I cannot raise my score, but I hope that I will be able to review the paper at a future conference because I think the idea is great and needs to be explored further.

---

> > > > > ### Comment · Reviewer_dmp5 · 2024-12-02
> > > > > **Some more related work**
> > > > >
> > > > > I just stumbled upon two papers, which should be discussed (and potentially compared against) in a future version of your paper:
> > > > >
> > > > > 1. PROGRESS: Progressive Reinforcement-Learning-Based Surrogate Selection
> > > > >     Hess, Wagner and Bischl
> > > > >     International Conference on Learning and Intelligent Optimization, 2013 [Link](https://link.springer.com/chapter/10.1007/978-3-642-44973-4_13)
> > > > > 2. Evolutionary Model Type Selection for Global Surrogate Modeling
> > > > >     Gorissen, Dhaea, De Turck
> > > > >     JMLR 2009 [Link](https://jmlr.csail.mit.edu/papers/volume10/gorissen09a/gorissen09a.pdf)
> > > > >
> > > > > Nonetheless, I think it is great that you pick up on this line of work, and encourage you to pursue it even further.

---

### Meta-Review · Area_Chair_aDCW · 2024-12-31

**Metareview:**

This work considers using an ensemble of surrogate models (including from multiple model classes, i.e., GPs, Gradient Boosting, Random Forests, Extremely randomized Trees).  Weights are updated online based on their performance with respect to new observations.  While much of the prior work on ensembling for BO considers the case of meta-learning (where models from previous tasks were used), or ensembling of acquisition function strategies, this work is somewhat unique in its approach.

Reviewers commented on the clarity of the paper, and found the algorithm to be of interest.  Although dynamically weighting ensembles of models has been considered extensively in previous works (see e.g., some of the works of Feurer and Wistuba et al.), few works have considered ensembling models of only the target task itself.  Reviewers also found the number of benchmark problems to be convincing.

However, there were still a number of gaps highlighted by the reviewers that were not addressed during the rebuttal.  This includes some potentially very serious issues with the experimental setup—not updating the surrogate models in a consistent/sequential fashion (dmp5), lack of theoretical discussion behind weighting scheme and consideration of GP-free baselines (2TAw), and missing baselines of more standard ensembling / model averaging techniques (mBTA).  Largely, reviewers indicated that the work was not ready for publication prior to the review, and did not think that the authors adequately addressed their concerns in the rebuttal.

I encourage the authors to carefully consider and execute on the detailed suggestions from the reviewers, and resubmit to a future conference.

**Additional Comments On Reviewer Discussion:**

The reviewers identified significant issues with the experimental setup and lack of comparison against simpler baselines.  The authors did not adequately respond to these critiques.

---

### Decision · Program_Chairs · 2025-01-22

Reject